# The real-time infection hospitalisation and fatality risk across the COVID-19 pandemic in England

Thomas Ward [1] ✉, Martyn Fyles[1], Alex Glaser[1], Robert S. Paton[1], William Ferguson [1] & Christopher E. Overton[1,2]

The COVID-19 pandemic led to 231,841 deaths and 940,243 hospitalisations in England, by the end of March 2023. This paper calculates the real-time infection hospitalisation risk (IHR) and infection fatality risk (IFR) using the Office for National Statistics Coronavirus Infection Survey (ONS CIS) and the Real-time Assessment of Community Transmission Survey between November 2020 to March 2023. The IHR and the IFR in England peaked in January 2021 at 3.39% (95% Credible Intervals (CrI): 2.79, 3.97) and 0.97% (95% CrI: 0.62, 1.36), respectively. After this time, there was a rapid decline in the severity from infection, with the lowest estimated IHR of 0.32% (95% CrI: 0.27, 0.39) in December 2022 and IFR of 0.06% (95% CrI: 0.04, 0.08) in April 2022. We found infection severity to vary more markedly between regions early in the pandemic however, the absolute heterogeneity has since reduced. The risk from infection of SARS-CoV-2 has changed substantially throughout the COVID-19 pandemic with a decline of 86.03% (80.86, 89.35) and 89.67% (80.18, 93.93) in the IHR and IFR, respectively, since early 2021. From April 2022 until March 2023, the end of the ONS CIS study, we found fluctuating patterns in the severity of infection with the resumption of more normative mixing, resurgent epidemic waves, patterns of waning immunity, and emerging variants that have shown signs of convergent evolution.

The COVID-19 pandemic has been attributed to 6.91 million mortalities globally, up to the 31st March 2023[1]. This has had far reaching implications for public health policy worldwide and led to unprecedented interventions. The clinical severity observed in response to an infection with SARS-CoV-2 has evolved over time as a consequence of emerging variants of concern, vaccination campaigns, high infection attack rates, and changes to the clinical management of patients.

Emerging variants of concern have been the impetus behind resurgent waves of SARS-CoV-2 incidence and changes to the severity profile of infections. The Alpha variant was first sequenced in September 2020 and became the dominant variant in the UK. Relative to wild type, Alpha[2] was estimated to have a 62% (Hazard Ratio (HR) – 1.62

(95% CI: 1.48, 1.78)) and 73% (HR – 1.73 (95% CI: 1.41, 2.13)) increased risk of hospitalisation and mortality, respectively. The discernible replacement of Alpha by Delta was detected in early 2021[3] and it became the dominant variant by June[4]. Delta was found to have a substantially increased risk of hospitalisation relative to Alpha with a HR of 1.85 (95% CI: 1.39, 2.47)[5]. Delta was replaced by the Omicron BA.1 in December 2021[6] with increased vaccine escape noted as a significant factor. Omicron BA.1 was estimated to have an almost threefold reduction in the risk of hospital admission relative to Delta[7]. In March 2022, Omicron BA.2 replaced Omicron BA.1[8], and there was found to be limited evidence of a difference in the severity of infection[7]. Omicron BA.5 replaced Omicron BA.2, in June 2022, as the dominant variant in the

[1]UK Health Security Agency, Data, Analytics and Surveillance, Nobel House, London SW1P 3JR, UK. [2]University of Liverpool, Department of Mathematical Sciences, Peach Street, Liverpool, UK. ✉e-mail: Tom.Ward@UKHSA.gov.uk

UK[9]. There was evidence that Omicron BA.5[10] infections may be associated with an increased risk of hospitalisation relative to Omicron BA.2. Following a summer epidemic wave of SARS-CoV-2 infections in 2022, Omicron further diversified. Several of these lineages convergently acquired mutations on the receptor binding domain that are associated with immune evasion[11]. These lineages include BF.7 (a BA.5.2 derivative), BA.5.3 sub lineages BQ.1 and BQ.1.1, as well as lineages derived from BA.2.75. Notably, a BA.2 recombinant carrying many of these mutations (XBB) drove a wave of incidence in Singapore and later became dominant in the UK[12].

The vaccination campaign in the United Kingdom began on the 8th December 2020[13]. The campaign was implemented in phases with the groups prioritised by clinical risk. Phase 1 included 9 high priority groups and began with care home residents, individuals over the age of 70, the clinically extremely vulnerable, frontline healthcare staff and social care workers[14]. The remaining phase 1 groups included those aged 50 to 69 years old. Subsequently phase 2[15] was implemented in April 2021 and began a further age stratified approach beginning with those aged 40–49 and concluding with the 18-29 age group. Phase 3, 4, and 5 focused on booster campaigns, the clinically vulnerable, and children over the age of 12.

The primary vaccinations administered in the UK were the AstraZeneca vaccine, Vaxzevria, and the Pfizer vaccine, Comirnaty. The impact of vaccination campaigns on the disease severity in the population has been influenced by the timing of the campaign and the variant specific response. An early study found that Vaxzevria had a vaccine efficacy of 66.7% (95% Confidence Interval (CI): 57.4, 74.0), 14 days after the second dose[16] for wild type. The efficacy of the vaccine was estimated to be 81.3% (95% CI: 60.3, 91.2) for individuals that had a longer prime-boost interval. The vaccine efficacy of 2 doses of Vaxzevria against symptomatic infection was estimated to be 70.4% (95% CI: 43.6, 84.5)[17] for Alpha. It was later estimated that the vaccine effectiveness for Delta was 67.0% (95% CI: 61.3, 71.8)[18] and limited protection against symptomatic disease was found for Omicron BA.1[19]. There was no significant difference found in the vaccine effectiveness for Omicron BA.1 and Omicron BA.2[20] and there was limited evidence for the effectiveness of two doses of Vaxzevria for Omicron BA.4/BA.5. Vaxzevria was the primary vaccine administered to those aged over 40 in the UK after concerns of haemostatic complications in younger ages. Comirnaty was administered at the start of the vaccination campaign and subsequently it was primarily administered to those aged under 40. The vaccine efficacy for two doses of Comirnaty was estimated to be 96.2% (95% CI: 93.3, 98.1)[21] early in the pandemic. Later studies estimated the vaccine effectiveness to be 89.5% (95% CI: 85.9, 92.3) for Alpha[22] and 88.0% (95% CI: 85.3, 90.1) for Delta[18]. Evidence of the two dose effectiveness of Comirnaty for Omicron subvariants[23] was limited with wide uncertainty[24]. For the 3rd and 4th booster vaccinations, the Joint Committee on Vaccination and Immunisation stated[25] a preference for Comirnaty or a half dose of Spikevax and where mRNA vaccines could not be used then individuals were offered Vaxzevria.

On the 15th August 2022, the Medicines and Healthcare products Regulatory Agency (MHRA) approved the use of a bivalent COVID-19 vaccine made by Moderna, which targeted both the 2020 SARS-CoV-2 viral strain and Omicron BA.1[26]. The Pfizer/BioNTech bivalent vaccine was approved by the MHRA, less than a month later, on the 3rd September 2022[27], which also targeted the 2020 SARS-CoV-2 viral strain and Omicron BA.1. A second bivalent vaccine from Pfizer/BioNTech was approved by the MHRA in November 2022, which targeted Omicron BA.4/BA.5 and the 2020 viral strain. A study published near the end of 2023[28] found that the vaccine effectiveness of bivalent BA.1 boosters against hospitalisation peaked at 53.0% (95% CI: 47.9, 57.5) 2 to 4 weeks after a dose was administered and at 10 weeks this had reduced to 35.9% (95% CI: 31.4, 40.1). In September 2023 the MHRA approved Pfizer/BioNTech[29] and Moderna's[30] bivalent vaccine to target Omicron XBB.1.5; with ongoing work to understand this vaccine's effectiveness against emerging variants. Analysis of vaccine effectiveness through population-based studies in the UK has been impacted by the cessation of free testing in the UK. This limits the understanding of variant prevalence and impacts the means to adjust for past infection in statistical analyses, with limited information on the infection ascertainment rates.

Improvements in the medical management of patients infected with COVID-19 has reduced the hospitalisation and fatality risk for those infected with the virus. Research found the use of non-invasive continuous positive airways pressure and awake prone positioning to be associated with improved patient outcomes[31]. To try and reduce the risk of severe disease in the clinically extremely vulnerable, anti-viral medicine, and neutralising monoclonal antibodies have been made available in the community and within Secondary Care[32]. This has included nirmatrelvir and ritonavir (Paxlovid), sotrovimab (Xevudy), remdesivir (Veklury), and molnupiravir (Lagevrio)[33].

This paper describes changes over time to the real-time infection hospitalisation risk (IHR) and infection fatality risk (IFR) using the Office for National Statistics Coronavirus Infection Survey (ONS CIS) and Real-time Assessment of Community Transmission (REACT) prevalence survey. We assess the impact by region and age over the length of the pandemic.

## Results

### Parameter estimation

The estimated PCR positivity length for every dominant variant across the epidemic in the UK can be seen in Supplementary Fig. 1 and Supplementary Table 1. The temporal changes in the time from symptom onset to hospitalisation and death by age groups can be seen in Supplementary Figs. 2 and 3, respectively. The PCR test sensitivity modelling for every dominant variant and each age group can be seen in Supplementary Fig. 4. The modelling estimates for the time from symptom onset to a first positive test by age and region can be seen in Supplementary Figs. 5–9.

### Real-time infection hospitalisation risk – national

The IHR in England peaked at 3.39% (95% Credible Intervals (CrI): 2.79, 3.97) in January 2021, during the period when the Alpha variant was dominant and most of the population were unvaccinated (Fig. 1). After the rollout of the vaccination programme, the IHR started declining rapidly. Near the end of the Delta period, in November 2021, the IHR had reduced to 0.58% (95% CrI: 0.50, 0.67) and the lowest IHR was estimated to be 0.32% (95% CrI: 0.27, 0.39) in December 2022. Since this time, the IHR has fluctuated and it was estimated to be 0.47% (95% CrI: 0.39, 0.59) by February 2023. Overall, the IHR has declined by 86.03% (80.86, 89.35) since January 2021. The REACT and ONS CIS prevalence estimates and hospital admissions attributed to COVID-19 can be seen in Supplementary Figs. 10 and 11, respectively.

### Real-time infection hospitalisation risk – age groups

The IHR peaked in January and February 2021 for the age groups over 44 (Table 1 and Fig. 2). The IHR peaked later for those aged between 6 to 24 (March 2021) and 25 to 44 (April 2021). The IHR declined in every age group from May 2021 and reached the lowest estimated value in April 2022 for the age groups over 54 and in December 2022 for the age groups under 55. Since this time, we have seen fluctuations in the estimated IHR for every age group. Since the peak in early 2021 until February 2023 the IHR has decreased by 92.51% (88.84, 94.52) for the ≥ 75 age group; 92.25% (88.04, 94.63) for the 65 to 74 age group; 91.90% (87.86, 94.37) for the 55 to 64 age group; 91.51% (87.90, 94.00) for the 45 to 54 age group; 92.72% (89.50, 94.89) for the 25 to 44 age group; and 88.30% (80.48, 92.25) for the 6 to 24 age group. The REACT and ONS CIS prevalence estimates and hospital admissions attributed to COVID-19 for each age group can be seen in Supplementary Figs. 12 and 13, respectively. The full results for each prevalence study and age group can be seen in Supplementary Figs. 14–19.

## Real-time infection hospitalisation risk – regions

Following the national estimates, we estimated the highest IHR in all NHS regions was in January and February 2021 (Fig. 3 and Table 2). After early 2021, the IHR rapidly declined in all NHS regions. The IHR reached the lowest estimated value in March and April 2022 in the East of England, London, South East, and the North East and Yorkshire when Omicron BA.2 was dominant. The IHR continued declining until October 2022 in the North West and until December 2022 in the Midlands and South West. All regions have since seen a fluctuating pattern in the IHR. The REACT and ONS CIS prevalence estimates and hospital admissions attributed to COVID-19 for each NHS region can be seen in Supplementary Figs. 20 and 21, respectively. The full results for each study and region can be seen in Supplementary Figs. 22 to 28.

## Real-time infection fatality risk – national

The infection fatality risk in England peaked at 0.97% (95% CrI: 0.62, 1.36) in January 2021, after which time the IFR began to rapidly decline (Fig. 4). In November 2021, at the end of the Delta period, the IFR had reduced to 0.11% (95% CrI: 0.08, 0.15). The IFR continued to decrease through the Omicron BA.1 and Omicron BA.2 period reaching 0.06% (95% CrI: 0.04, 0.08) in April 2022. Since this time, we have observed fluctuations in the IFR and at the end of Februrary 2023 it was estimated to be 0.10% (95% CrI: 0.07, 0.16). Since the peak in January 2021 the IFR had declined, overall, by 89.67% (80.18, 93.93) up to February 2023. The national REACT and ONS CIS prevalence estimates and

deaths attributed to COVID-19 can be seen in Supplementary Figs. 10 and 29.

## Real-time infection fatality risk – age groups

In every age group we estimated the highest IFRs to be in January and February 2021 (Table 3 and Fig. 5). For most of the age groups the IFR reached the lowest estimated value in March and April 2022 with the exception of the 6 to 24 age group which reached its lowest estimated

**Table 1 | Key Estimates of the Infection Hospitalisation Risk by Age**

| Infection Hospitalisation Risk (%) – Age Groups | | | |
|---|---|---|---|
| Age Groups | Highest IHR (95% CrI) | Lowest IHR (95% CrI) | Latest IHR (95% CrI) |
| 6–24 | 0.36 (0.25, 0.46) | 0.03 (0.03, 0.04) | 0.04 (0.03, 0.06) |
| 25–44 | 1.39 (1.10, 1.73) | 0.07 (0.05, 0.08) | 0.10 (0.08, 0.13) |
| 45–54 | 2.18 (1.77, 2.64) | 0.11 (0.09, 0.14) | 0.19 (0.14, 0.25) |
| 55–64 | 4.34 (3.39, 5.36) | 0.23 (0.19, 0.29) | 0.35 (0.26, 0.48) |
| 65–74 | 9.97 (7.85, 12.15) | 0.48 (0.39, 0.61) | 0.77 (0.56, 1.08) |
| ≥ 75 | 33.93 (27.16, 39.42) | 1.86 (1.50, 2.35) | 2.53 (1.9, 3.44) |

The highest, lowest, and the latest median posterior estimates of Infection Hospitalisation Risk by age group across the epidemic in England, with 95% credible intervals.

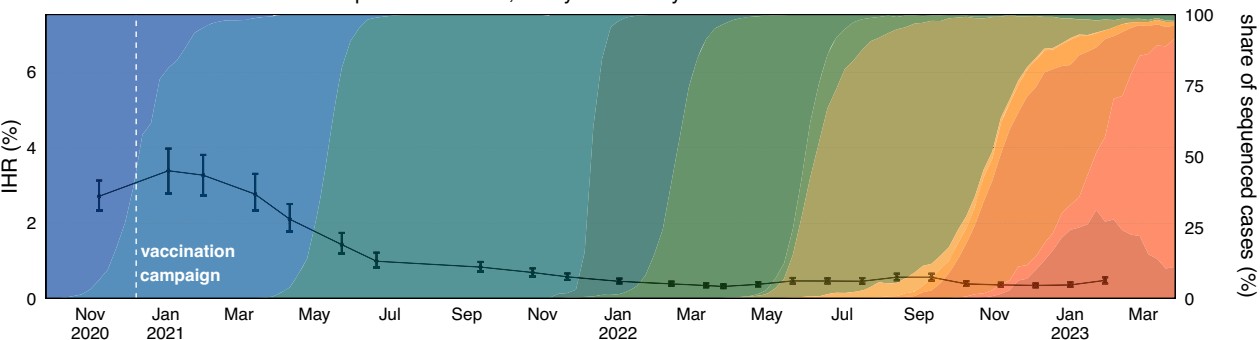

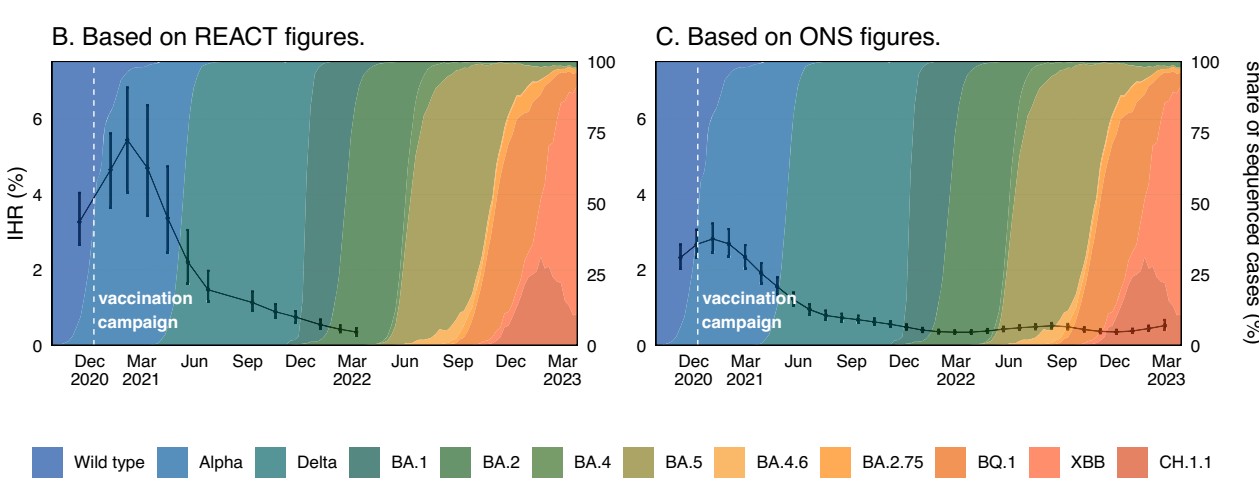

**Fig. 1 | The Infection Hospitalisation Risk for England. A** The posterior estimates of the median infection hospitalisation risk for England, based on the combined REACT and ONS sampling, with 95% credible intervals. **B** Posterior estimates of the median infection hospitalisation risk for England, based on REACT sampling, with

95% credible intervals. **C** Posterior estimates of the median infection hospitalisation risk for England, based on ONS sampling, with 95% credible intervals. Not all estimates derived from the ONS CIS study have been plotted. The data for the figure are provided as a Source Data file.

IHRs over time for age groups with 95% CrI
ONS / REACT studies combined pre–March 2022, ONS study plotted afterwards

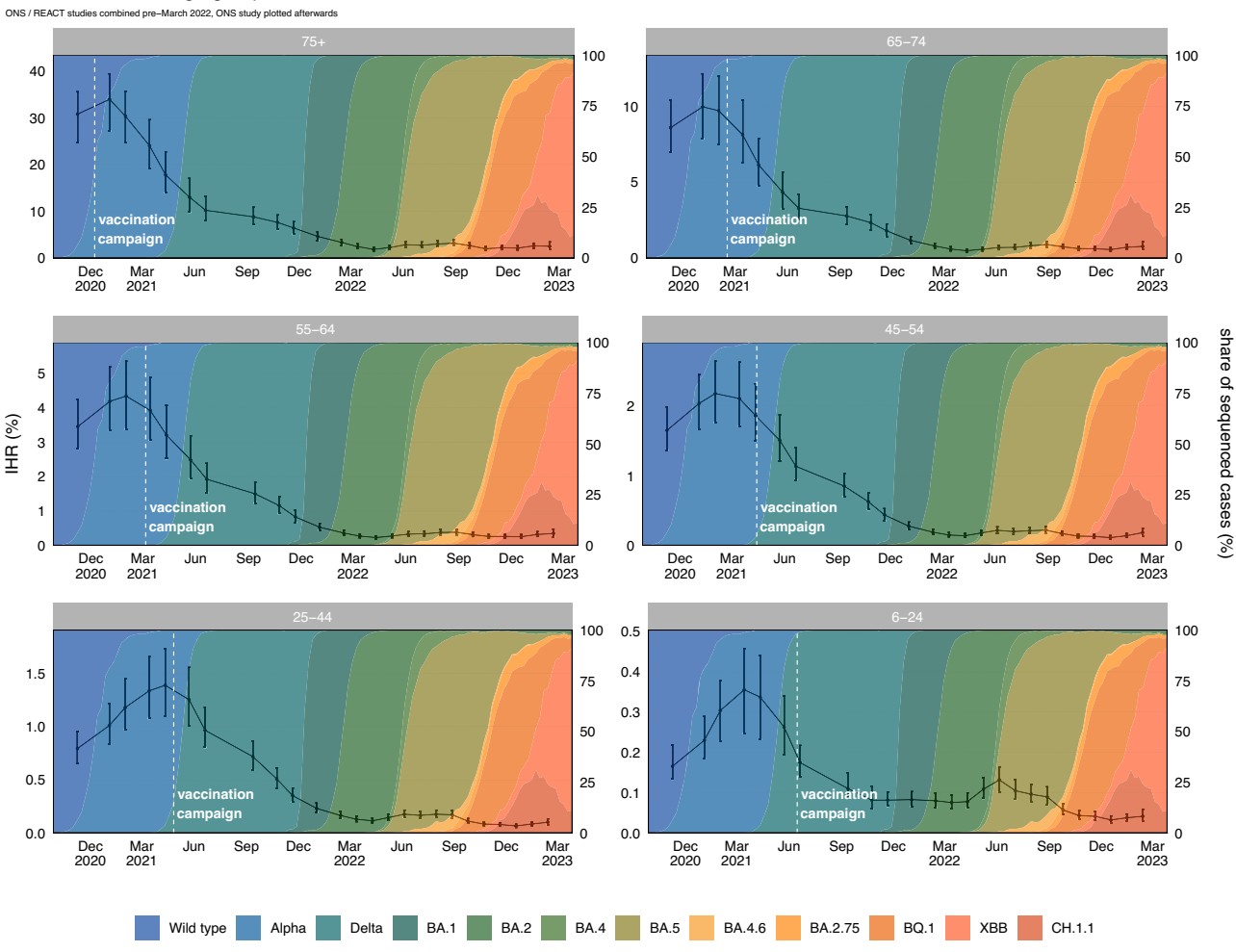

**Fig. 2 | The Infection Hospitalisation Risk by Age.** The posterior estimates of the median infection hospitalisation risk by age group, based on the combined REACT and ONS sampling, with 95% credible intervals. The data for the figure are provided as a Source Data file.

value in February 2023. From early 2021 until February 2023 we have seen a decline in the IFR of 95.95% (91.95, 97.70) for the ≥ 75 age group; 95.19% (91.00, 97.18) for the 65 to 74 age group; 94.19% (88.93, 96.71) for the 55 to 64 age group; 92.92% (87.74, 95.72) for the 45 to 54 age group; 90.71% (82.53, 95.05) for the 25 to 44 age group; and 92.74% (78.17, 98.35) for the 6 to 24 age group. The REACT and ONS CIS prevalence estimates and deaths attributed to COVID-19 for each age group can be seen in Supplementary Figs. 12 and 30, respectively. The full results for each study and age group can be seen in Supplementary Figs. 31–36.

**Real-time infection fatality risk – regions**
Similar to the trends seen in the IHR, we found the highest estimated IFR for most regions to be in January 2021 with the exception of London and the North East that peaked in November 2020 (Table 4 and Fig. 6). The IFR reached the lowest estimated value in March and April 2022 for every English region except the South West, South East, and East of England (estimated to be in July 2022). We subsequently have observed fluctuations in the estimated IFR for every region. The REACT and ONS CIS prevalence estimates and deaths attributed to COVID-19 for each region can be seen in Supplementary Figs. 20 and 37, respectively. The full results for each study and region can be seen in Supplementary Figs. 38 to 46. The regional age composition and index of multiple deprivation scores can be seen in Supplementary Fig. 47.

## Discussion
Over the course of the pandemic in England, the severity from infection of SARS-CoV-2 has substantially decreased. Changes to the IHR and IFR have been driven by a combination of vaccination, immunity from infection, patient management, and the demographic distribution of infections. We observe that since the January 2021 peak until February 2023, there has been a decline of 86.03% (80.86, 89.35) and 89.67% (80.18, 93.93) in the IHR and IFR, respectively. The early decline, since January 2021, was likely a consequence of the rollout of the vaccination programme, which reached the oldest and most vulnerable individuals in December 2020 and January 2021. Consequently, we observe a later peak in March and April 2021 in the IHR for the age groups under 45. We observed considerable regional heterogeneity at the start of the pandemic, which has substantially reduced post vaccination and following high infection attack rates in the population. Nationally, the IHR and IFR continued to decline until December and April 2022, respectively, which followed Autumn and Winter booster campaigns in England. However, the period following early 2022 has been characterised by an undulatory pattern in the IFR and IHR in response to the timing of vaccination campaigns, resurgent epidemic waves, and emerging variants. We estimated by the end of the study that 4.73 (3.85, 5.93) individuals in 1,000 that are infected with SARS-CoV-2 will be hospitalised and that 1.00 (0.67, 1.56) individual in 1,000 that are infected will die.

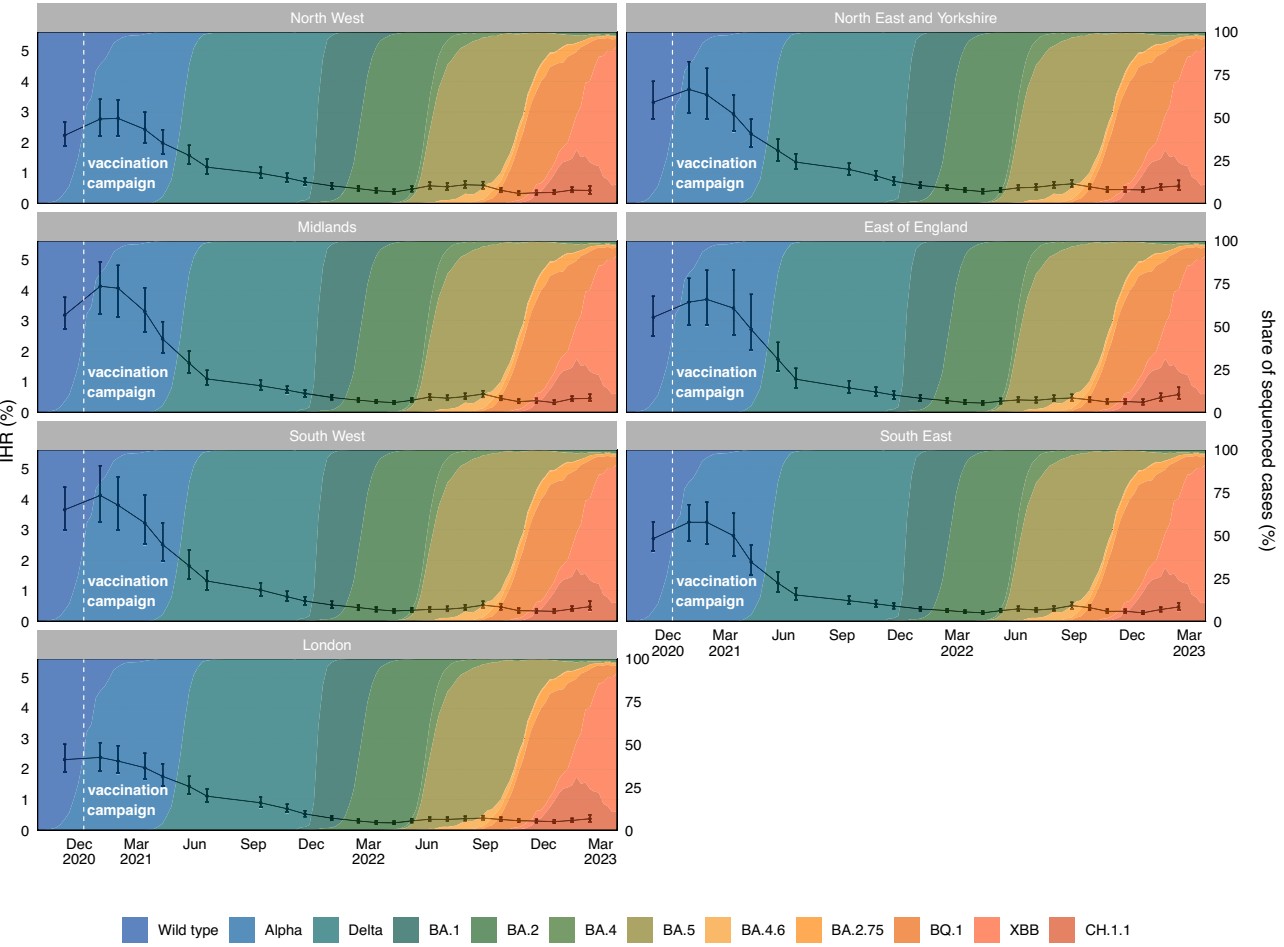

**Fig. 3 | The Infection Hospitalisation Risk for the Regions of England.** The posterior estimates of the median infection hospitalisation risk for the regions of England, based on the combined REACT and ONS sampling, with 95% credible intervals. The data for the figure are provided as a Source Data file.

### Table 2 | Key Estimates of the Infection Hospitalisation Risk for the Regions of England

| Infection Hospitalisation Risk (%) – Regions | | | |
|---|---|---|---|
| Regions | Highest IHR (95% CrI) | Lowest IHR (95% CrI) | Latest IHR (95% CrI) |
| East of England | 3.70 (2.85, 4.65) | 0.30 (0.24, 0.39) | 0.59 (0.43, 0.82) |
| London | 2.39 (1.96, 2.87) | 0.24 (0.19, 0.29) | 0.38 (0.28, 0.50) |
| Midlands | 4.13 (3.23, 4.93) | 0.32 (0.27, 0.41) | 0.46 (0.37, 0.59) |
| North East and Yorkshire | 3.74 (2.96, 4.65) | 0.39 (0.33, 0.47) | 0.58 (0.44, 0.76) |
| North West | 2.79 (2.20, 3.39) | 0.33 (0.27, 0.41) | 0.43 (0.33, 0.57) |
| South East | 3.25 (2.63, 3.82) | 0.28 (0.23, 0.34) | 0.47 (0.36, 0.61) |
| South West | 4.12 (3.25, 5.1) | 0.33 (0.27, 0.40) | 0.48 (0.36, 0.66) |

The highest, lowest, and the latest median posterior estimates of Infection Hospitalisation Risk by English region across the epidemic, with 95% credible intervals.

Early point estimates of the IFR in 2020, calculated from antibody surveys, ranged from 1.15% (95% Prediction Interval Range (PI): 0.78%, 1.79%) in high income countries to 0.23% (95% PI: 0.14%, 0.42%) in low income countries[34]. Further IFR estimates retrospectively calculated, of the largely pre-vaccination period in the UK, have ranged from 1.57% (95% Uncertainty Interval (UI): 1.22, 2.47) in April 2020 to 1.20% (95% UI: 0.88%, 1.73%) in January 2021[35]. The study period for this paper began on the 8th November 2020 and therefore does not cover the early pandemic. However, we did not find a reduction in the IFR until after January 2021. Nonetheless, the considerable uncertainty in these estimates overlap with the credible intervals of this study's estimates in

January 2021. Some of these early serological studies were not adequately powered, with regards to sample size, and draw from existing surveys[34–37] that may not be representative of the general population.

To calculate incidence and the time to a clinical event we used temporally variable parameters. The time from symptom onset date (used here as a proxy for infection date) to hospitalisation and death evolves in response to epidemic phases[38], changes to clinical management, prior immunity, and the pathogenesis of novel variants[39]. The length of the infectious period of a randomised sampled cohort changes in response to epidemic phases. PCR positivity was found to vary across the variants that became dominant in the UK. We found the

IFRs over time for England with 95% CrI

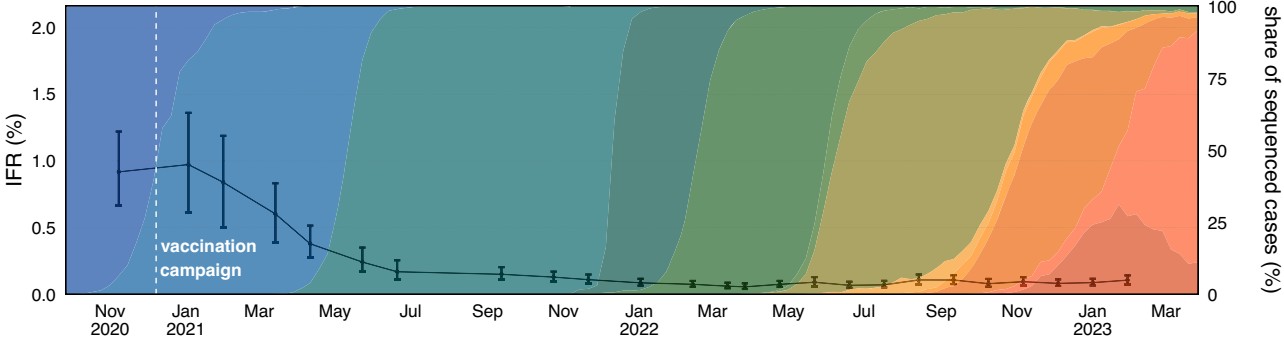

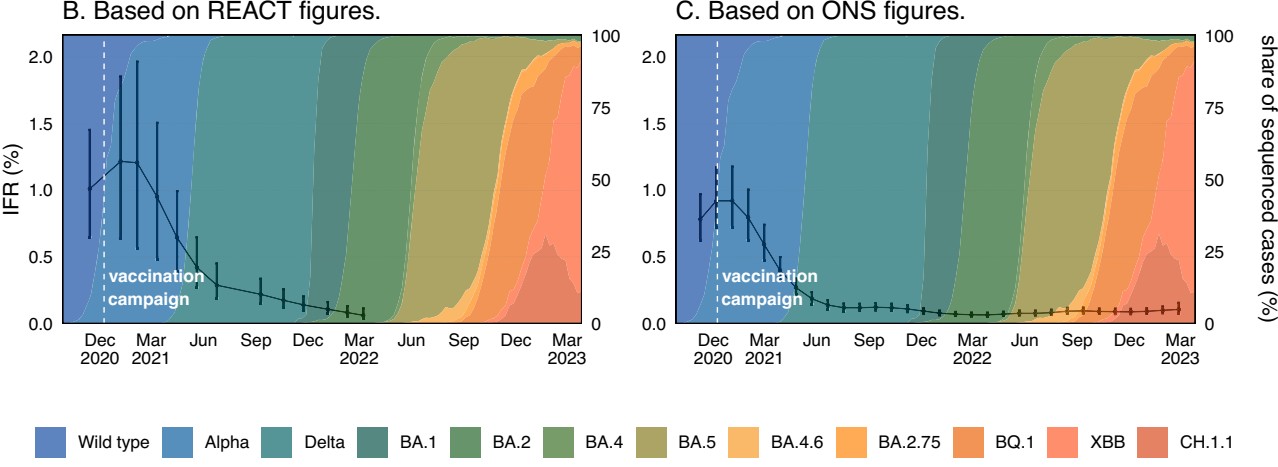

| Wild type | Alpha | Delta | BA.1 | BA.2 | BA.4 | BA.5 | BA.4.6 | BA.2.75 | BQ.1 | XBB | CH.1.1 |

**Fig. 4 | The Infection Fatality Risk for England. A** The posterior estimates of the median infection fatality risk for England, based on the combined REACT and ONS sampling, with 95% credible intervals. **B** Posterior estimates of the median infection fatality risk for England, based on REACT sampling, with 95% credible intervals. **C** Posterior estimates of the median infection fatality risk for England, based on ONS sampling, with 95% credible intervals. Not all estimates derived from the ONS CIS study have been plotted. The data for the figure are provided as a Source Data file.

### Table 3 | Key Estimates of the Infection Fatality Risk by Age

| Infection Fatality Risk (%) – Age Groups | | | |
| --- | --- | --- | --- |
| **Age Groups** | **Highest IFR (95% CrI)** | **Lowest IFR (95% CrI)** | **Latest IFR (95% CrI)** |
| **6–24** | 0.0022 (0.0015, 0.0036) | 0.00016 (0.00004, 0.00042) | 0.00016 (0.00004, 0.00042) |
| **25–44** | 0.040 (0.029, 0.055) | 0.0024 (0.0017, 0.0037) | 0.0037 (0.0022, 0.0063) |
| **45–54** | 0.20 (0.14, 0.27) | 0.0091 (0.0066, 0.014) | 0.014 (0.0093, 0.022) |
| **55–64** | 0.71 (0.49, 1.00) | 0.023 (0.016, 0.038) | 0.041 (0.026, 0.068) |
| **65–74** | 2.84 (1.96, 3.89) | 0.064 (0.046, 0.10) | 0.14 (0.089, 0.22) |
| **≥ 75** | 16.95 (11.36, 22.76) | 0.52 (0.36, 0.81) | 0.68 (0.43, 1.12) |

The highest, lowest, and the latest median posterior estimates of Infection Fatality Risk by age group across the epidemic in England, with 95% credible intervals.

Alpha variant to have the longest PCR positivity and a reduction in length was estimated for the Omicron variants.

The criteria for a hospital admission or mortality attributed to COVID-19 can be multifaceted, obfuscated by the comorbidities, clinical policy, and extrinsic factors including hospital pressure. The absolute value of the IHR and IFR estimates are sensitive to the criteria used to define a hospitalisation or mortality from COVID-19. The definition commonly used in the pandemic within the UK has been 28-day deaths[40], which was thought to be likely an underestimate of true deaths from COVID-19[41]. However, the 60-day deaths definition could overestimate COVID-19 deaths in some subgroups, particularly in older individuals who have higher baseline mortality rates. Death certificate confirmed COVID-19 deaths suffered from changes to death reporting practices across the pandemic[40]. Determining the cause of a hospital admission from surveillance data requires assumptions without further clinical information. Hospital surveillance data can include some nosocomial patients as well as patients admitted to hospital for other illnesses who tested positive for COVID-19 on admission. Although the absolute values of the IHR and IFR are sensitive to these criteria, the temporal trends are robust, provided the definitions remain consistent over time.

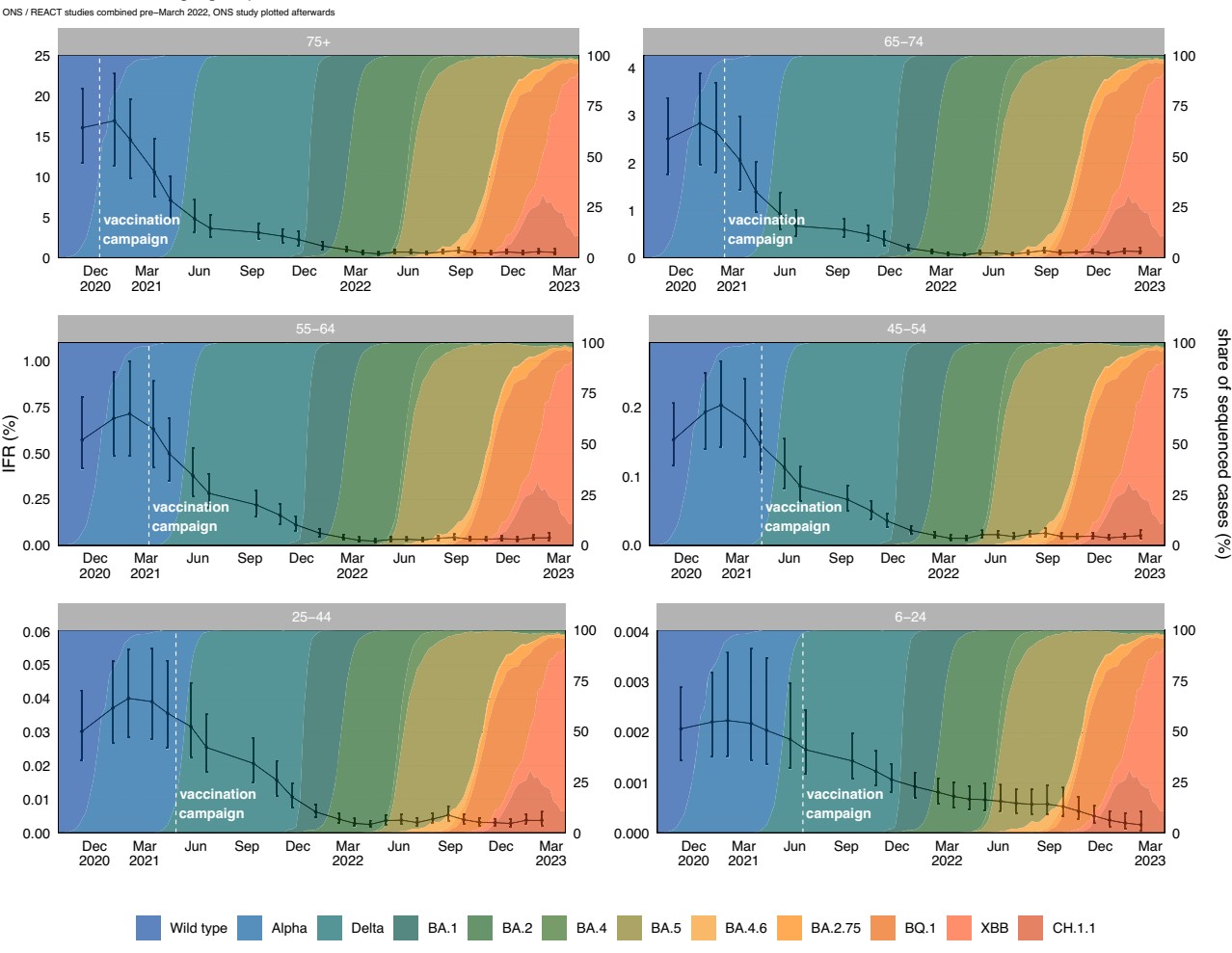

**Fig. 5 | The Infection Fatality Risk by Age.** The posterior estimates of the median infection fatality risk by age group, based on the combined REACT and ONS sampling, with 95% credible intervals. The data for the figure are provided as a Source Data file.

### Table 4 | Key Estimates of the Infection Fatality Risk for the Regions of England

| Infection Fatality Risk (%) – Regions | | | |
| --- | --- | --- | --- |
| Regions | Highest IFR (95% CrI) | Lowest IFR (95% CrI) | Latest IFR (95% CrI) |
| East Midlands | 1.35 (0.80, 1.83) | 0.05 (0.04, 0.09) | 0.11 (0.07, 0.17) |
| East of England | 1.55 (0.98, 2.03) | 0.05 (0.04, 0.09) | 0.11 (0.08, 0.17) |
| London | 0.61 (0.43, 0.84) | 0.03 (0.02, 0.05) | 0.05 (0.03, 0.08) |
| North East | 0.90 (0.66, 1.21) | 0.07 (0.05, 0.10) | 0.12 (0.08, 0.19) |
| North West | 0.98 (0.63, 1.34) | 0.06 (0.04, 0.09) | 0.10 (0.07, 0.16) |
| South East | 1.43 (0.95, 1.85) | 0.05 (0.04, 0.07) | 0.09 (0.062, 0.14) |
| South West | 1.38 (0.79, 1.9) | 0.06 (0.04, 0.09) | 0.11 (0.07, 0.17) |
| West Midlands | 1.23 (0.82, 1.68) | 0.06 (0.04, 0.09) | 0.10 (0.07, 0.16) |
| Yorkshire and The Humber | 0.95 (0.62, 1.34) | 0.06 (0.05, 0.09) | 0.11 (0.07, 0.17) |

The highest, lowest, and the latest median posterior estimates of the Infection Fatality Risk by English region across the epidemic, with 95% credible intervals.

The method used in this paper calculates the proportion of infections that led to deaths and hospital admissions attributed to COVID-19. Since we are interested in the IFR/IHR across grouped time-periods, or rounds, we made an approximation to the method that relies on the assumption that the risk is constant within each round. This assumption simplifies the method, without substantially affecting the within round estimates. This method is slightly affected by the epidemic phase severity bias[42], whereby severity is overestimated during phases of growth and underestimated during phases of decline.

To adjust for this bias, we would need to adjust for different incubation period distributions conditional on patient outcomes. That is, we would need to construct two prevalence time-series, prevalence among individuals that will get admitted to hospital or die and prevalence among individuals that will not. These two time-series would need to be deconvolved using different incubation periods, corresponding to the outcome, to estimate the bias-corrected infection incidence time-series. To obtain these two mutually exclusive prevalence time-series, data linking prevalence to patient outcome and

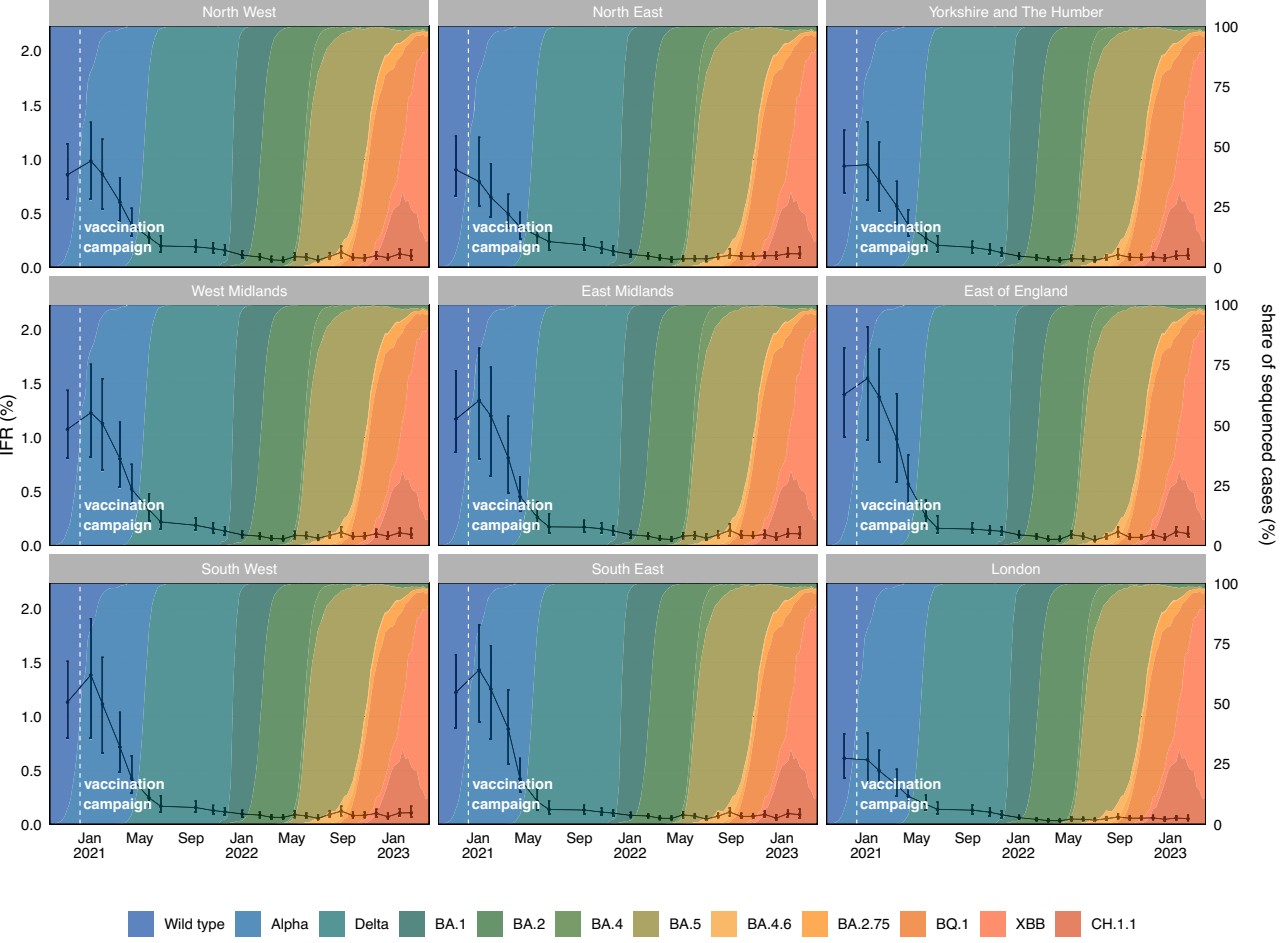

**Fig. 6 | The Infection Fatality Risk for the Regions of England.** The posterior estimates of the median infection fatality risk for the regions of England, based on the combined REACT and ONS sampling, with 95% credible intervals. The data for the figure are provided as a Source Data file.

exposure date is needed. The REACT survey data, available to this study, did not include the personally identifiable information needed to link to patient outcomes and this was not possible with ONS CIS due to the conditions of the participant consent agreement.

In this paper we have assessed temporal changes to the real-time infection hospitalisation and fatality risk from COVID-19. There has been considerable regional heterogeneity, which is likely a consequence of differing infection attack rates, distinct age compositions, and the relative differences in deprivation across England. At the end of February 2023, the IHR and IFR in England are estimated to be 0.47% (95% CrI: 0.39%, 0.59%) and 0.10% (95% CrI: 0.067%, 0.16%), respectively.

## Methods
This section describes the methods used to calculate the real-time IFR and IHR from the REACT and ONS CIS studies. We discuss the calculations for the viral parameter history, which includes calculating the length of PCR positivity by variant, PCR test sensitivity by variant, and the time delay modelling from symptom onset to a first positive test, hospital admission and death. For each study (Supplementary Fig. 48) we describe the modelling to calculate prevalence, incidence, and infection severity risk.

### Epidemiological data
There were two large surveys in the United Kingdom that provided real-time estimates of SARS-CoV-2 positivity: the ONS Coronavirus

Infection Survey[43] and the REACT 1 antigen survey coordinated by Imperial College London in conjunction with Ipsos MORI[44]. ONS data was sourced through the Secure Research Service (SRS)[45] where we extracted demographic and regional breakdowns of the Reverse Transcription Polymerase Chain Reaction (RT-PCR) test results. Non-identifiable aggregated REACT study data was provided through a data agreement between UKHSA and Imperial College London.

REACT 1 was a repeat cross-sectional study that estimated SARS-CoV-2 prevalence in England[46] from May 2020 until March 2022. The study aimed to sample between 95,000 to 175,000 individuals randomly for each survey round over the age of 5[47], which was updated from 100,000 to 150,000 individuals in the original study protocol[46]. The study sent out recruitment letters on a 4–6-week basis to a randomised sample from the National Health Service patient register that aimed to be nationally representative. For children under the age of 18 the recruitment letters were sent to a parent or guardian. Individuals that chose to participate were sent throat and nasal swabs, which were sent for RT-PCR testing. The participants were then invited to complete an online questionnaire, which included demographics, infection history, and behavioural topics. Overall, the REACT 1 and REACT 2 studies found a response rate of 23.4% across the study period[48]. The REACT reports and study protocol have been published through Imperial College London and Welcome Open Research[46,49]. Supplementary Figs. 49–51 describes the sample sizes for each age group and region over time.

The ONS CIS study was produced by the ONS in collaboration with the Wellcome Trust, University of Oxford, IQVIA, Lighthouse Laboratories, Joint Biosecurity Centre, UKHSA and the University of Manchester[50]. The study began in April 2020 as a pilot and invited 20,000 households from the ongoing Labour Force Survey[51] and those that had agreed previously to participate in the Opinions and Lifestyle Survey[52]. Then in August 2020, the survey expanded to invite a randomised household sample from AddressBase[53]. The extended household study aimed to achieve around 150,000 swabs a fortnight in England between October 2020 and March 2023. The study requested that the entire household (over the age of 2) take a nose and throat swab, which was sent to the Lighthouse Laboratory for RT-PCR testing. Tests were conducted by home visits from a study worker and in April 2022 a proportion of participants of the study were asked to post their samples. At this time the number of swabs required reduced by 25% with an aim to swab 227,300 individuals every 28 days in England. From the 1st August 2022, the study collections had moved to being entirely remote. The ONS reported[50] an attrition rate of 0.62% in December 2020 that fluctuated between a high of 1.37% in July 2021 to the lowest rate of 0.32% in December 2021. The ONS study paused at the end of March 2023 with the intention to restart later in the year. Supplementary Figs. 49–51 describes the sample sizes for each age group and region over time.

Mortality data, subset by age and geography, were sourced from the UKHSA COVID-19 death linelist. To limit capturing deaths that were less likely to be linked to a COVID-19 infection we only included deaths that had occurred 60 days following a positive RT-PCR test. Hospitalisation attributed to an infection with COVID-19 were collected from the NHSE&I situational report data[54]. Mortality data was available for the 9 English regions[55] and hospital data was reported for the 7 NHS English Health regions[56].

Population size data, stratified either by age group or region, were sourced from the ONS at a yearly resolution. This data excludes communal establishments, as these were not sampled during the REACT or ONS surveys. Linear interpolation was used to provide estimates of population sizes for the midpoint of each round for each of the prevalence studies.

In order to provide context for the estimated temporal changes in IHR and IFR, we extracted data on sequenced cases from the Second Generation Surveillance System at the UKHSA, appending metadata on the age and residential region for the case. To complement the national, regional and age stratified analyses we calculated the proportion of sequenced cases associated with wild type, Alpha, Delta, and Omicron lineages.

Due to data protection regulations the model methodologies were developed to work across different data platforms including the ONS SRS, the UKHSA Halo system and Public Health England data infrastructure. The study period of this paper is from the 8th November 2020 until March 2023.

## Methodology outline

Our methodology for calculating the infection severity risk features two key steps. Firstly, we must take the number of positive and negative tests for each survey round and estimate the number of new infections for that round, referred to as the incidence for that round. This requires us to adjust for several quantities: positivity duration, delay from infection to testing positive, test sensitivity and test specificity. Given an estimate of the round incidence, we then produce an estimate of how many clinical outcomes, such as deaths or hospitalisations, are attributed to a given round. This is achieved by estimating the delay from symptom onset to clinical outcome, and then temporally adjusting clinical outcomes. Furthermore, many of these key parameters are known to vary over time, variant type, and age group. We first detail the models used to estimate these key

disease history parameters. Then, we will outline our final methodology for calculating the rate of severe outcomes, first using each prevalence study in isolation, then combining the results of the two studies.

A Bayesian methodology is used throughout, with all statistical models implemented using CmdStanR (version 0.6.1)[57], and posterior sampling performed using Hamiltonian Markov Chain Monte Carlo (MCMC). For each model we run 4 chains, each with 1000 warmup iterations followed by 1000 sampling iterations. Convergence was assessed using the $\hat{R}$ statistic[58], with convergence declared if $\hat{R} < 1.01$.

**Propagation of uncertainty.** The data used in this paper is stored in several different research environments, therefore it is not possible for this method to be implemented in a single Stan Bayesian program. Consequently, to propagate the uncertainty it is necessary to pass the posterior estimates from the models that estimate key disease parameters to later models that estimate the incidence or calculate the rate of severe outcomes. We have not made this explicit in the following methods and instead summarise our approach here.

If a model $\mathcal{M}_1$ depends on a parameter θ that is obtained from model $\mathcal{M}_0$, then this is provided to model $\mathcal{M}_1$ via the prior

$$\theta \sim \mathcal{D}(\hat{\mu_\theta}, \hat{\sigma_\theta}),$$

where $\mathcal{D}$ is an appropriate parametric distribution. For example, a beta distribution is an appropriate choice to describe the posterior distribution of parameters bounded between 0 and 1. The values of $\hat{\mu_\theta}, \hat{\sigma_\theta}$ were obtained by using maximum likelihood to fit $\mathcal{D}$ to the posterior draws of θ from model $\mathcal{M}_0$. For the parameters that do need to be provided between models, we found that either normal or beta distributions were able to provide satisfactory fits for the parameters.

## Disease history parameter estimation

**Positivity duration.** From the ONS COVID Infection Survey, we have longitudinal testing data for individuals. After initially entering the survey, individuals test weekly for the first four weeks, before testing monthly for the remainder of their time on the survey. We estimate the duration of positivity $\tau_{pos} \in \mathbb{R}_+$, defined as the delay from when a case first becomes positive to when a case ceases to be positive, by decomposing into two delays such that $\tau_{pos} = \tau_{fp} + \tau_{EoP}$, where $\tau_{fp} \in \mathbb{R}_+$ is the delay from symptom onset to first testing positive, and $\tau_{EoP} \in \mathbb{R}_+$ is the delay from when the case first tests positive to when the case ceases to test positive. As part of this, we make the assumption that time of symptom onset approximates the time at which the case becomes positive[59]. While it would be possible to not make this assumption and use an interval censoring model to estimate the delay from the infection becoming positive to the infection testing positive, due to the size of the intervals relative to the delay, the uncertainty on any estimates produced using this approach would be too large and would consequently degrade results (please see a schematic in Supplementary Fig. 52).

**Delay from first positive test, to the end of positivity.** For each case, data can be obtained on two delays: the delay from the first positive test to the last positive test followed by a negative test $\tau_{lp} \in \mathbb{R}_+$, and the delay from the first positive test to the first negative test $\tau_{fn} \in \mathbb{R}_+$. Therefore, we have that $\tau_{EoP} \in \left[\tau_{lp}, \tau_{fn}\right]$.

It is likely that the test positivity duration has changed with the emergence of different variants, therefore we condition $\tau_{EoP}$ upon the dominant variant at the time of the case's infection. We also make the assumption that the positive duration of each case is distributed according to a Weibull distribution, and for each variant we perform

interval-censored regression to estimate the positive duration distribution. This is achieved by defining a date range, over which individuals who first test positive within this range are assumed to be infected with that variant. The date ranges used are:

$$t \leq 2020 - 11 - 07 \sim \text{Wildtype},$$

$$2020 - 11 - 08 \leq t \leq 2021 - 05 - 08 \sim \text{Alpha},$$

$$2021 - 05 - 09 \leq t \leq 2021 - 12 - 18 \sim \text{Delta},$$

$$2021 - 12 - 19 \leq t \leq 2022 - 03 - 06 \sim \text{BA.1},$$

$$2022 - 03 - 07 \leq t \leq 2022 - 05 - 22 \sim \text{BA.2},$$

$$2022 - 05 - 23 \leq t \leq 2022 - 10 - 08 \sim \text{BA.4/5},$$

$$2022 - 10 - 09 \leq t \leq 2023 - 03 - 31 \sim \text{BQ/CH/XBB}.$$

Letting $\tau_{\text{EoP}}^{(i)}$ denote the delay for the $i^{th}$ case, we assume that $\tau_{\text{EoP}}^{(i)} \sim \text{Weibull}(\alpha, \lambda_v)$. Here, the shape parameter $\alpha \in \mathbb{R}_+$ is shared across all variants with the rate $\lambda_v \in \mathbb{R}_+$ parameter conditional upon the variant assigned to the $i^{th}$ case, where $v \in \{1, \ldots, 7\}$ denotes which variant the $i^{th}$ case is assigned to.

For the $i^{th}$ case, we must compute the likelihood for the event

$$\tau_{\text{EoP}} \in \left[ \tau_{\text{lp}}, \tau_{\text{fn}} \right] \quad (1)$$

which has a likelihood given by

$$\mathbb{P}\left( \tau_{\text{EoP}} \in \left[ \tau_{\text{lp}}, \tau_{\text{fn}} \right] | \alpha, \lambda_v \right) = F_{\text{weibull}}\left( \tau_{\text{fn}}; \alpha, \lambda_v \right) - F_{\text{weibull}}\left( \tau_{\text{lp}}; \alpha, \lambda_v \right). \quad (2)$$

For our priors, we let $\alpha \sim \text{Exponential}(0.1)$, and $\lambda_v \sim \mathcal{N}(15, 10)$.

Our assumption that the shape parameter is shared across all variants is due to the presence of large censoring intervals that make inferring the shape of the distribution difficult. Consequently, we find it necessary to use data from multiple variants to infer the shape of the Weibull distribution.

**Onset to first positive test delay.** For symptomatic individuals with a positive COVID-19 test, the ONS COVID Infection Survey reports symptom onset date. This allows data to be extracted on symptom onset and first positive test time for each patient. The mean time from symptom onset to a positive test is used as a proxy measure to estimate the temporal variation in PCR positivity by approximating the average time from becoming positive to testing positive.

This distribution is likely to vary significantly compared to the community COVID-19 testing, because community testing is based on healthcare seeking behaviour among the general population as opposed to randomised testing within the population. This delay is further affected by epidemic phases, whereby during times of growth the observed delays are shorter, and during times of decay the observed delays are longer, consequently it is necessary for the parameters of this delay to be modelled as time-varying. After visualising the observed delays, we find that the skew-normal distribution is the only positive unbounded continuous distribution that would be appropriate to model this distribution that is available in Stan. Other distributions available in Stan were either symmetric,

featured heavy tails or had other undesirable properties that meant that they were not appropriate for modelling the data.

Let $\tau_{\text{fp}}^{(i)} \in \mathbb{R}$ be the delay from the $i^{th}$ case developing symptoms to first testing positive. Under our assumptions we have that

$$\tau_{\text{fp}}^{(i)} \sim \text{SkewNormal}(\xi_k, \omega_k, \upsilon_k), \quad (3)$$

where $\underline{\xi} \in \mathbb{R}^K, \underline{\omega} \in \mathbb{R}_+^K$ and $\underline{\upsilon} \in \mathbb{R}^K$ are the location, scale, and shape parameters of the skew normal distribution respectively, and $k \in \{1, 2, \ldots, K\}$ denotes which survey round the $i^{th}$ observation belongs to. The parameters of the skew normal are modelled using first order random walk smoothing priors, which enforce smoothness by assuming that increments of the random walk are normally distributed, i.e. let $\underline{x} = (x_1, \ldots, x_K) \in \mathbb{R}^K$ be a first order random walk ($RW1$), then we have that $x_{i+1} - x_i \sim \mathcal{N}(0, \sigma^2)$ where $\sigma \in \mathbb{R}_+$ is a hyperparameter to be estimated that controls the smoothness of the random walk. Hence, for modelling the parameters of the skew normal, we let

$$\underline{\xi} \sim RW1(\sigma_\xi)$$

$$\log(\underline{\omega}) \sim RW1(\sigma_\omega)$$

$$\underline{\upsilon} \sim RW1(\sigma_\upsilon)$$

where $\sigma_\xi, \sigma_\omega, \sigma_\upsilon \sim \mathcal{N}_+(0, 1)$.

A survey round is defined as the sampling period that was determined by the REACT (typically 2 weeks) and the ONS CIS study.

**Onset to clinical event delay.** To measure the distribution of delays from onset to clinical event, we use the Secondary Uses Service (SUS)[60] data from the NHS and the UKHSA death line list data sets. These data are daily censored, so we consider this as doubly-interval censored data. In these data, we only observe patients conditional on the clinical event occurring, which introduces right-truncation, since data are only observed before the final day of data collection $T$, which in this case is 23rd April 2023.

To estimate the mean delay from onset (as reported by patients) to hospitalisation or death, we fit to the data using a Weibull and lognormal distribution respectively, with both models accounting for interval censoring and right truncation. These distributions were selected as the best performing distributions out of the gamma, Weibull, and lognormal distributions, according to the Pareto-smoothed importance sampling leave-one-out cross-validation scores[61]. We fit the models to data aggregated into three-month periods by symptom onset date, in order to obtain time-varying delay parameters. The method here adapts the methods from Ward & Johnsen[38], Ward et al.[62], and Vekaria, et al.[63].

In this method, we assume that symptom onset time $S \in \mathbb{Z}$ for each individual sits within an interval $[s_1, s_2]$, where $s_1$ is the reported symptom onset date and $s_2$ is the day after, i.e., $s_2 = s_1 + 1$. Similarly, the clinical event time $E \in \mathbb{Z}$ sits within an interval $[e_1, e_2]$ where $e_2 = e_1 + 1$. The likelihood of observing a given clinical event time, conditional on the observed onset interval, is given by:

$$\mathbb{P}(E \in [e_1, e_2] | S \in [s_1, s_2], E < T) = \frac{\mathbb{P}(E \in [e_1, e_2], S \in [s_1, s_2])}{\mathbb{P}(E < T, S \in [s_1, s_2])}. \quad (4)$$

This likelihood could be modelled by numerically integrating across the observation intervals. However, this would be very computationally expensive. Instead, we can include estimated event times for each patient as latent variables within our model[62], which we assume to be uniformly distributed across the observation interval. Introducing

these latent variables $e^*$ and $s^*$, our likelihood function simplifies to

$$
\begin{aligned}
\mathbb{P}\left(E = e^*|S = s^*, E < T\right) &= \frac{\mathbb{P}\left(E = e^*, S = s^*\right)}{\mathbb{P}\left(E < T, S = s^*\right)} \\
&= \frac{\mathbb{P}\left(E = e^*|S = s^*\right)\mathbb{P}\left(S = s^*\right)}{\mathbb{P}\left(E < T|S = s^*\right)\mathbb{P}\left(S = s^*\right)} \\
&= \frac{\mathbb{P}\left(E = e^*|S = s^*\right)}{\mathbb{P}\left(E < T|S = s^*\right)} \\
&= \frac{f_\theta\left(e^* - s^*\right)}{F_\theta\left(T - s^*\right)}
\end{aligned}
\tag{5}
$$

where $f_\theta(.)$ is the probability density function of the parametric distributions with parameters $\theta_1$ and $\theta_2$. We combine this likelihood with prior distributions for our latent variables given by

$$e^* \sim \text{Uniform}(e_1, e_2),$$

$$s^* \sim \text{Uniform}(s_1, s_2).$$

We assume $\theta_1$ represents the mean for admissions and the log mean for mortalities, and follows a weakly informative normal prior distribution. For the delay to hospitalisation, we assume

$$\theta_1 \sim \mathcal{N}(10, 5).$$

For the delay to death, we assume

$$\theta_1 \sim \mathcal{N}(\log(27.5), 0.5).$$

We assume $\theta_2$ represents the shape parameter for admissions, and has the prior

$$\theta_2 \sim \text{Exponential}(0.0001).$$

For mortalities, we assume $\theta_2$ represents the log of the standard deviation, and has the prior

$$\theta_2 \sim \mathcal{N}(0, 1).$$

This model is fit using MCMC implemented in Stan, with full model formula

$$e^* \sim \text{Uniform}(e_1, e_2),$$

$$s^* \sim \text{Uniform}(s_1, s_2),$$

$$\text{loglikelihood} \sim \log(f_\theta(e^* - s^*)) - \log(F_\theta(T - s^*)).
\tag{6}$$

We consider time varying onset to clinical outcome delays, fitting the delays independently to each three-month time period, starting from September 2020.

**Sensitivity and specificity.** The REACT and ONS studies both use RT-PCR tests that can have variable sensitivity and specificity, which were not adjusted for in the reported results of either study. These values are influenced by swabbing protocol, laboratory, specimen storage, days since symptom onset, site of swab, age, and variant mutations. Primers are adjusted if a drop in sensitivity is observed for a variant[64,65]. In addition, given that test sensitivity is conditional upon days since symptom onset, it is known that the average test sensitivity will be affected by epidemic phase bias[66].

The average RT-PCR test sensitivity for individuals in each round is calculated from two key components: an estimate of the test sensitivity as a function of the time to symptom onset, and an estimated delay distribution of the time between symptom onset and first positive test. Given the existence of potential differences in viral dynamics between different variants, we estimate a different test sensitivity trajectory for each variant, and we estimate the time from symptom onset to testing positive for each round and each study.

To calculate RT-PCR test sensitivity, we have assessed repeat tests by age group from ONS CIS where an individual must have a symptom onset date, with at least one positive test up to 12 days prior and 30 days after symptom onset date.

To fit the RT-PCR test sensitivity, we adapt the method of Binny et al.[67] who fit a piecewise linear logistic regression model, with the binary outcome of a positive or negative test, using days relative to symptom onset date, $d \in \mathbb{R}$, as the primary explanatory variable.

$$p_{\text{sens}}(d) = \text{logit}^{-1}\left(\beta_0 + (d - D)(\beta_1 + (\beta_2 - \beta_1)\mathbb{1}_{\{d > D\}})\right)
\tag{7}$$

In practice, we found a piecewise linear logistic regression to be a poor fit for our data, and as such we instead modify this function by: removing the changepoint and replacing with a sigmoid, which results in a smoothed out changepoint that is more biologically plausible; and replacing the piecewise linear terms with piecewise polynomial terms, which allows for greater flexibility in fitting to the data. The derived function form is given by:

$$
\begin{aligned}
p_{\text{sens}}(d) = \text{logit}^{-1}\Big(&\beta_0 + \beta_1 |d - D|_1^{\lambda_1}\left(1 - \Phi((d - D + 2)/2)\right) \\
&+ \beta_2 |d - D|_1^{\lambda_2}\Phi((d - D - 2)/2)\Big),
\end{aligned}
\tag{8}
$$

where $\beta_1, \lambda_1, \lambda_2 \in \mathbb{R}_+$ and $\beta_0, D \in \mathbb{R}$, $\beta_2 \in \mathbb{R}_-$. We use $\Phi$ to denote the cumulative distribution function of the standard normal distribution, however for this purpose we are using it as a sigmoidal function rather than for its probabilistic interpretation. The following priors are used in fitting this function:

$$
\begin{aligned}
\beta_0 &\sim \mathcal{N}(0, 5), \\
\beta_1 &\sim \mathcal{N}_+(0, 1), \\
\beta_2 &\sim \mathcal{N}_-(0, 1), \\
\lambda_1, \lambda_2 &\sim \mathcal{N}_+(0, 1), \\
T &\sim \mathcal{N}(0, 5).
\end{aligned}
$$

**Infection hospitalisation and fatality risk modelling**
The ONS and REACT studies reported positive testing rates over time to help inform the public health response to the pandemic via calculations of the effective reproduction number and acting as inputs into government modelling. With results at both a national level as well as geographic and demographic subdivisions, it is possible to detect higher risk areas in need of greater intervention. By combining the estimated incidence with clinical outcome data in the form of hospital admissions and mortalities, the IHR and IFR can be calculated by the method set out below.

**Estimating prevalence from positivity.** For each round $k \in [1, \ldots, K]$ and subgroup (i.e., region or age group) indexed according to $s \in [1, \ldots, S]$, we calculate the expected test sensitivity $p_{\text{avg\_sens}}$ for that round using

$$
\begin{aligned}
p_{\text{avg\_sens}}(k, s) &= E_t\left[p_{\text{sens}}(t, k, s)\right] \\
&= \int_{-12}^{30} p_{\text{sens}}(t, k, s) \cdot f_{SN}(t|\xi_{k,s}, \alpha_{k,s}, \omega_{k,s})\partial t,
\end{aligned}
\tag{9}
$$

where $p_{\text{sens}}(t, k, s)$ is our estimate of the probability of testing positive $t$ days after symptom onset in the $k^{\text{th}}$ round for the $s^{\text{th}}$ subgroup given in equation (8), $f_{SN}$ is the probability density function of the skew normal distribution that we use to model the delay from symptom onset to testing for positive tests, and $\xi_{k,s}, \alpha_{k,s}, \omega_{k,s}$ are the estimated

parameters of the skew normal distribution for the $k^{th}$ estimate of positivity and the $s^{th}$ subgroup.

Test specificity $p_{spec} \in [0,1]$, does not have sufficient data available to produce an estimate. We apply a strong prior that encodes prior beliefs that RT-PCR tests are highly sensitivity, with a false positive rate of approximately 1 in 10,000, i.e., $p_{spec} \sim \mathcal{B}eta(10000,1)$.

We assume that the prevalence is constant across each round $p_{prev} \in [0,1]$. Given $p_{prev}$, $p_{spec}$, and $p_{avg\_sens}$, the probability that a randomly tested individual will test positive $p_{pos} \in [0,1]$, is given by

$$p_{pos} = p_{prev} \cdot p_{avg\_sens} + \left(1 - p_{prev}\right)\left(1 - p_{spec}\right) \quad (10)$$

Let $N_{k,s} \in \mathbb{Z}_+$ be the number of tests performed for a given round and stratum, and $P_{k,s} \in [0, N_{k,s}]$ be the number of tests that were positive for that round and stratum. Then the likelihood is given by

$$P_k \sim \mathrm{Binomial}\left(N_k, p_{pos}\right) \quad (11)$$

A hierarchal model structure with second order random walk smooths and random effects is used when estimating $p_{prev}(k,s)$. To maintain identifiability of the model in the presence of both smoothing and random effects, which are effectively two different smooths at the round-subgroup level, we use a special formulation adapted from a BYM2 framework[68] given by

$$p_{prev}(k,s) = \mathrm{logit}^{-1}\left(\beta_k + \gamma\left(\sqrt{\alpha_1}f_{avg\_prev}(k) + \sqrt{\alpha_2}f_{prev}(k,s) + \sqrt{\alpha_3}\xi(k,s)\right)\right) \quad (12)$$

where $f_{avg\_prev}$ and $f_{prev}$ are logit-scaled smooth functions, and $\xi \sim \mathcal{N}(0,1)$ are the random effects, $\beta_k \in \mathbb{R}$ the intercept term, $\gamma \in \mathbb{R}_+$ an overall scale term, and $\alpha_1, \alpha_2, \alpha_3 \in [0,1]$ the elements of a 2-simplex, i.e. $\alpha_1 + \alpha_2 + \alpha_3 = 1$.

Both $f_{avg\_prev}$ and $f_{prev}(\cdot, s)$ are constrained to have a mean of zero to maintain identifiability. In addition, we ensure that both $f_{avg\_prev}$ and $f_{prev}(\cdot, s)$ are on approximately the same scale as the random effects by placing a standard normal distribution prior on them $f_{avg\_prev}f_{prev} \sim \mathcal{N}(0,1)$, in addition to their improper smoothing prior. Therefore, the overall scale is controlled by $\gamma$ given that

$$\mathrm{Var}\left[\sqrt{\alpha_1}f_{avg\_prev}(k) + \sqrt{\alpha_2}f_{prev}(k,s) + \sqrt{\alpha_3}\xi(k,s)\right] = \alpha_1 + \alpha_2 + \alpha_3 = 1, \quad (13)$$

which results in a well-identified model structure. For the random walk smoothing priors we let

$$f_{avg\_prev} \sim \mathrm{RW2}\left(\sigma_{avg\_prev}\right),$$

$$f_{prev}(\cdot, s) \sim RW2\left(\sigma_{prev}\right), \text{for } s \in 1:S,$$

$$\sigma_{avg\_prev}, \sigma_{prev} \sim \mathrm{Exponential}(100)$$

where $RW2(\sigma)$ implies a penalty on the second order derivative in the form of

$$\frac{d^2}{dt^2}f(k) \approx f(k+1) - 2f(k) + f(k-1) \sim \mathrm{N}(0,\sigma), \forall k. \quad (14)$$

The $\alpha$ terms control the relative contribution to the variance from each of the components, and we let $\alpha \sim \mathrm{Dirichlet}(2,2,2)$.

In addition to the prevalence for each age group, we also estimated the national prevalence by Multilevel Regression and Poststratification, which allows us to perform statistical adjustment for demographics that are over/under represented in the sample.

The above method for calculating the prevalence in each age group uses a multilevel regression approach, and it remains to perform a poststratification step to estimate the national prevalence by reweighting the prevalence for each age group. Letting $p_{prev}^{nat}(s)$ be the poststratified estimate of national prevalence, calculated as

$$p_{prev}^{nat}(s) = \frac{\sum_{k=1}^{K} p_{prev}(k,s) \cdot N_k}{\sum_{k=1}^{K} N_k} \quad (15)$$

where $N_k$ is the population of the $k^{th}$ strata. We poststratified our results according to the age breakdown of our sample, on the basis that age is the most important variable to account for when producing nationally representative estimates of the IHR.

**Calculating incidence attributed to round.** For each survey (REACT and ONS), we converted the estimated prevalence rates $p_{prev}(k,s)$ for population stratum $s$ and round $k$ into an incidence time series $I(k,s)$ using this expression:

$$I_s(k) = \frac{p_{prev}(k,s)\Omega(k,s)l(k)}{\tau_{pos}(s)}. \quad (16)$$

Here, $\tau_{pos}(s)$ is the expected duration for which an individual tests positive, $\Omega(k,s)$ is the population size of stratum $s$ during round $k$, and $l(k) \in \mathbb{N}$ is the length of the round in days. We note that $\tau_{pos}$, and other parameters used in calculating it, are derived from the ONS study, since REACT surveying did not provide adequate data to estimate these values.

One way to think of this equation is that initial positive test frequencies are shifted back in time to more accurately reflect when individuals were infected. In this paper, we shift the testing dates to symptom onset date rather than infection date, since we have more reliable data on the delay distributions post symptom onset date. Finally, we multiply by the population $\Omega$ to scale up our sample to population wide numbers, but also divide by the time for which someone tests positive $\tau_{pos}$.

**Calculating outcome counts attributed to round.** Given an estimate of the number of new infections that occurred during a round, it remains to estimate the number of clinical outcomes attributed to individuals infected during that round, which then finally allows us the calculate the rate out severe outcomes.

There is a time delay between symptom onset and clinical outcome[38], which must be accounted for in the relationship between incidence and hospitalisation or death[69-71]. For a given stratum $s$, we must establish a time series $d_s(t)$ that models clinical outcomes $c_s(t)$ in that stratum by date of symptom onset rather than by date of outcome. We model $d_s(t)$ as

$$d_s(t) = \sum_{t'=0}^{\infty} c_s(t+t') \cdot p_s(t'|t) \quad (17)$$

where $p_s(t',|,t)$ is the probability that the time from symptom onset to outcome is $t'$ for someone in stratum $s$, given that they were infected on day $t$. This approximates the method for mapping outcomes to date of symptom onset[69], under the assumption that each round has a constant risk.

From the daily-level time series $d_s(t)$, it remains to estimate the number of clinical events attributed to the $k^{th}$ round in stratum $s$, denoted by $D_s(k)$, using

$$D_s(k) = \sum_{i \in \mathscr{I}(k)} d_s(i) \quad (18)$$

where $\mathscr{I}(k)$ is the set of timepoints that correspond to the $k^{\text{th}}$ round.

**Outcome risk modelling.** Given posterior draws of $I_s(k), D_s(k)$, we compute the clinical outcome rate, however the resulting estimates are noisy, implying that some smoothing is required.

We first fit a normal distribution to the posterior draws of $D_s(k), I_s(k)$ for each round and subgroup, as we are able to easily provide this parametric summary of the posterior as an input to the model. Letting $\mu_s(k), \sigma_s(k)$ be the parameters of the normal distributions for each round and stratum, we have that

$$D_s(k) \sim \mathcal{N}\big(\mu_s^{(D)}(k), \sigma_s^{(D)}(k)\big),$$

$$I_s(k) \sim \mathcal{N}\big(\mu_s^{(I)}(k), \sigma_s^{(I)}(k)\big) \qquad (19)$$

Employing the Jeffrey's prior for the clinical outcome rate, the posterior distribution for the infection risk $R_s(k) \in [0,1]$ is given by

$$R_s(k) \sim \mathcal{B}\mathrm{eta}\big(D_s(k) + 0.5, I_s(k) - D_s(k) + 0.5\big) \qquad (20)$$

In addition, we place a second order random walk smoothing prior, with random effects present on $R_k(t)$. A similar structure is used to the hierarchal model employed when estimating prevalence to maintain identifiability in the presence of both a random walk smooth and a random effect smooth;

$$R_k(t) = \mathrm{logit}^{-1}\Big(\beta + \gamma\big(\sqrt{\alpha}f_R(t) + \sqrt{1-\alpha}\xi(t)\big)\Big),$$

$$\beta \sim \mathcal{N}(-4,1),$$

$$\gamma \sim \mathcal{N}(0,2),$$

$$\alpha \sim \mathrm{Beta}(2,2),$$

$$f_R \sim RW2(\sigma_R) \; f_R \sim \mathcal{N}(0,1), \mathrm{mean}(f_R) \sim \mathcal{N}(0,0.001),$$

$$\sigma \sim \mathrm{Exponential}(100),$$

$$\xi \sim \mathcal{N}(0,1) \qquad (21)$$

**Infection study model combination**

To combine the prevalence studies, each study was matched temporally over the same sampling periods, using test results from the ONS study matched to REACT round sampling dates. We weight the model so that the two samples are assigned weights by adjusting for the relative sample sizes. This weighting method follows the approach of Balcome, et al.[72] adapting the work of Haddad et al.[73].

Here we let $D_O$ and $I_O$ denote $D_s(k)$ and $I_s(k)$, respectively, for the ONS infection survey study. In this vein, let $D_R$ and $I_R$ denote $D_s(k)$ and $I_s(k)$ for the REACT study. Letting $R$ denote $R_s(k)$ then the posterior distribution for this event probability can be modelled using a Beta distribution, i.e.

$$R \sim \mathcal{B}\mathrm{eta}(D, I - D), \qquad (22)$$

where $I$ is the number of infections, and $D$ is the corresponding number of events. The aim of this combination method is to obtain a weighting factor $\hat{\alpha}$ such that $D = D_O + \hat{\alpha}D_R$ and $I = I_O + \hat{\alpha}I_R$.

We weight the two studies based on their relative sample sizes, so that when the sample sizes are equal, both studies are assigned equal weight, and otherwise the largest study is assigned greater weight. That is, we set $\hat{\alpha} = \frac{N_R}{N_O}$, where $N_O$ is the sample size of the

ONS study and $N_R$ is the sample size of the REACT study. Including $\hat{\alpha}$ into the posterior distribution for the event probability, we obtain

$$R \sim \mathcal{B}\mathrm{eta}\big(D_O + \hat{\alpha}D_R, I_O - D_O + \hat{\alpha}(I_R - D_R)\big) \qquad (23)$$

As in the previous section, when calculating the clinical outcome rate for a single study, we provided parametric summaries of the posteriors of $I, D$ terms to the model as inputs. We also place the same smoothing prior on $R$ as in the previous section.

**Reporting summary**

Further information on research design is available in the Nature Portfolio Reporting Summary linked to this article.

## Data availability

The Office of National Statistics COVID Infection Survey (ONS CIS) data can be accessed through the Secure Research Service of the ONS. For all other datasets used in this study please contact the UKHSA. UKHSA operates a robust governance process for applying to access protected data that considers: the benefits and risks of how the data will be used. compliance with policy, regulatory and ethical obligations. data minimisation. how the confidentiality, integrity, and availability will be maintained. retention, archival, and disposal requirements. best practice for protecting data, including the application of 'privacy by design and by default', emerging privacy conserving technologies and contractual controls. Access to protected data is always strictly controlled using legally binding data sharing contracts. UKHSA welcomes data applications from organisations looking to use protected data for public health purposes. To request an application pack or discuss a request for UKHSA data you would like to submit, contact DataAccess@ukhsa.gov.uk. Source data are provided with this paper.

## Code availability

The model code can be made available on request to DataAccess@ukhsa.gov.uk.

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

## Author contributions

TW conceived and led the study. T.W., A.G., W.F., M.F., and C.E.O. developed the methods and code for the time delay models. T.W., M.F., C.E.O., R.S.P., and A.G. developed the data criteria. T.W., A.G., M.F., and R.S.P. wrote the visualisations code. T.W., A.G., and M.F. developed the parameter models. T.W., A.G., M.F., and, C.E.O., and M.F. developed the methods and code for the infection risk models. T.W., A.G., C.E.O., and M.F. wrote the original manuscript. T.W., M.F., A.G., and C.E.O. reviewed the manuscript. T.W., M.F., A.G., and C.E.O. wrote the revisions.

## Competing interests

The authors declares no competing interests.

## Ethical statement

UKHSA have an exemption under regulation 3 of section 251 of the National Health Service Act (2006) to allow identifiable patient information to be processed to diagnose, control, prevent, or recognise trends in, communicable diseases and other risks to public health.
