## [Peer Review File · Nature Communications]

The real-time infection hospitalisation and fatality risk across the COVID-19 pandemic in EnglandREVIEWER COMMENTS

Reviewer #1 (Remarks to the Author):

The manuscript: "The Real Time Infection Hospitalization and Fatality Risk across the COVID-19 Pandemic in the United Kingdom" is well written and provides an overview of the reduction of the Infection and hospitalization fatality rate in the United Kingdom using 2 independent databases. The Method is well described and the conclusions compatible with the results. I suggest some minor corrections:

1) PCR positivity data show a reduction of positivity time for BA1. The differences in days are small, not being statistically and clinically relevant. This data was not prioritized in the abstract, in the discussion or in the conclusion as the main result, my suggestion is to remove this result from the manuscript and present only the IFR and HFR data. The manuscript is already quite extensive with a long introduction, methodology and results section.

2) In the fourth paragraph of introduction: "The 40 earliest estimates found that the two doses of Vaxzevria reduced symptomatic infection by 82.4% 41 (95% CI: 62.7, 91.7) [12] for wild type, dropping to 74.5% (95% CI: 68.4, 79.4) [13] for Alpha, 67.0% (95% CI: 61.3, 71.8) [14] for Delta, and very limited protection was found for Omicron BA.1 [15]. Currently there appears to be no significant difference in the effect of vaccination for Omicron BA.1 and Omicron BA.2 [16] and no current evidence for the effectiveness of two doses of Vaxzevria for Omicron BA.5." Reference 15 only has VE for Delta and not for Omicron. Reference 16 has VE for Omicron BA.1 and not the comparison among BA1 and BA.2. And in the last sentence regarding the BA5 variant, the authors do not quote any reference. It is important to correct the references in the text relating to Vaxzevria.

Reviewer #2 (Remarks to the Author):

I think that this is a very important paper and it has relevance not just for the results but also for methods used to estimate the infection fatality and hospitalisation rates.

The manuscript is well written and the presentation of the results is clear. The introduction and discussion are both very clear and I had no major comments on these sections

The key to this method is the prevalence surveys in the UK, from which incidence is estimated, and linking to the hospitalisation and death data through the distributions of the time for infection to event. I am aware of other studies which estimated the IFR through fitting SEIR type models. Are there other approaches that could be discussed. The discussion of this paper only really discusses the results and the possible reasons for the changes - vaccination, variant, waning protection - and differences among the regions associated with age. I think that there should be a comparison with other estimates of IFR/IHR if there are any such estimates.

The methods are really very complex and I think that it would be a good idea to try and draw a diagram to outline what you are doing and the stages. You have got daily data on deaths/hospitalisations by date of event and you need to convert this to events among those infected by estimated date of infection. You use the onset to clinical delay distribution to facilitate this back calculation. This gives you the $D(t)$ estimates. You then have the two surveys of prevalence, and you combine them and convert these into incidence estimates. This gives the $I(t)$. You do this explanation in 102-106 but I think a visual display like some flow diagram would help.

I was not exactly sure but think that the distinct parts are analysed separately and not all included within a full Bayesian model. I may be wrong here but this is what lines 280-284 suggest to me. I can understand why you do this as the problem is complex but this needs to be clarified. I think that it would be possible to propagate the estimation errors through simulations even if the distinct parts were analysed separately.

Also I am not sure if the 5% point of the IFR/IHR simulations based upon the 5% point of the

infection estimates is truly the lower limit of a 95% credible interval in a full bayesian analysis.

In the explanation of the results much is made of the periods when the different variants were circulating suggesting that the changes in the IHR/IFR are associated with variant. While they are there is also going to be an impact of vaccine uptake and waning of protection – particularly for infection, less for serious events – which would have more of an impact on the denominator of the ratios

Title – this mention the UK but everything in the paper is about England – React was only in England while the ONS survey did have components in Wales, Northern, Ireland and Scotland. I assume you used the English component of the ONS survey. There is no strong reason to suppose that the IHR/IFR would be substantially different in the other nations of the UK but it is better to replace UK in the title with England.

Minor Points

L91

we only included deaths that had occurred 60 days following a positive RT-PCR test and COVID-19 on the death certificate

What is the time period for this work – Early on the pandemic when testing was limited there were many covid deaths with no PCR test. How have you compensated for this in the analysis?

L128

Does the study end on 15/05/22 or are there missing rows.

L269-276

It took me a while to work out what you were trying to do here so it would be really helpful to put the full details in the supplementary materials. As I understood you want to combine the two surveys at the same time point. You might consider $(D_O + D_R)/(I_O + I_R)$ as an estimate of theta The scaling factor is the value you multiply D_R by so that the estimates of I from both studies are the same and so give equal weight. If you didn't do this and I_O and I_R were a bit different then the larger one would dominate and there is no reason to suppose that one survey's estimate of I is better than the others.

L275

Aren't a and b the shape parameters of the beta distribution. What values did you use for them or did you have hyper priors

L280-284

I understand that you have to take into account the variability in the estimation of I_O and I_R and what you do is reasonable. Is this the 95% credible interval in the paper and graphs. I don't think that this can be a true credible interval – how do you know that the coverage is 95%.

L 292 the CrI limits are the wrong way round – twice

L288. The credible interval is not a lot wider than you would get with binomial sampling. You don't report the incidence estimates but there are 3500 admissions to hospital at the peak in January 2021 and an IHR of 3.9% suggests 89000 infections with a 95% CI 3.80 to 4.06 which is not much smaller than the reported interval. Given the amount of estimation and calculations I would have expected a wider range. I may be mistaken or the sample sizes just dominate. In the age groups there may be a bigger difference.

L305-306 For most in the 25-44 age group vaccination did not start until April 2021

L310-319. I thought that this was confusing as you are talking about an 90% increase in IHR from the nadir to now and then you mention a 90% decrease from the peak to now, I guess, as the time is not mentioned. I know that this comes about as the 90% decrease from the peak to nadir is 36 to 1.8 and the 90% increase is from 1.8 to 3.5. I suggest no mentioning the % increases in so much detail and just concentrating on the absolute changes.

L328-340

The regional estimates are very interesting as the intervals do not overlap and London seems to have a much lower IHR in Jan 2021. At the time there were a lot of reports about hospitals in London being overwhelmed

The IFR analysis for the regions shows that London has the lowest IFR also – so is this associated with it having a much younger population – Supplementary fig 40 shows 65% in London under 44 and a much smaller % are 75+ - I see you discuss this so nothing extra is needed

L400

I think that the PCR positivity results could also be in the supplementary material. In the order of the analysis you really need this before getting the IHR/IFR so I found it strange to see it at the

end of the results.

L537

I think that the ONS survey does collect personal data and it can be linked to administrative health data within a trusted safe haven.

Reviewer #3 (Remarks to the Author):

This paper describes the evolution of the risk of hospitalisation and death following SARS-CoV-2 infection during the COVID-19 pandemic in the UK. This important piece of work combines different aggregate data sets using a thorough statistical approach. It allows to describe the evolution of two key indicators during the epidemic. The impact of vaccination and the evolution of the virus are discussed.

However, we believe that some significant improvements need to be made before publication.

There are four points that we would like to highlight.

The methods section should be more explicit about the data used in the model and what they are used for. This should be clear to readers unfamiliar with the organisation of covid-19 surveillance in the UK. It is important to explain why the ONS and REACT data are original (compared to what is available in other countries) and why they made this study possible.

Still in Methods, the reader should be able to better understand the general organisation of the model (what are the different steps and what data are used) without going into the details of the Bayesian models.

The methods used are original and advanced. It would be interesting to discuss its advantages and limitations compared to methods based on individual monitoring of infected individuals (e.g. reference [5] or 10.1016/j.eclinm.2022.101455).

Finally, the English used in the article is quite difficult to read. This applies especially to the title and the discussion. For example, we were puzzled by the following sentences. "Therefore, informed by the time delay distribution analysis, to capture 99% of deaths we included up to 60 days post a positive test or with COVID-19 as the primary cause on the death certificate." " Death certificate confirmed COVID-19 deaths suffer from changes to death reporting practices across the pandemic."

Below are some more specific comments.

The period of time considered in the study is not clearly defined.

It might be interesting to express IR and IFR as cases per 100,000. This would make them easier to read and avoid confusion with the percentage increase or decrease.

Line 54: When did free testing ceased in the UK?

Line 84: ONS and REACT 1 should be introduced in more detail here. How are individuals included in these surveys ? Is there a health-seeking bias? Some information is later in the article, such as line 144, but should be grouped here.

Line 93: NHSE&I is not mentioned as a data source in the summary.

Line 113-115. Did you mean "The first positive date is considered to be the date of the first positive test. The last positive test followed by a negative test is considered as the date of end of infection".

Line 151: The term "round" has not been defined.

Fig 1: Would it be possible to sort the legend by chronological order of variants?

Fig 2: Point out that the Y-scales are different for each age group.

Line 396: If I understand correctly, this parameter is used as an input to the model. Therefore, these results should be presented before those of the model or reported in the appendix.

Line 417: These percentage changes could be presented without decimals. This would make them easier to read.

Line 418: is "1.2 (1.16, 1.28) in 200" a typo ?

Line 432: The wording gives the impression that vaccination is one of the causes of the increase!

Line 474: The wording gives the impression that vaccination has not significantly changed the IFR and IHR.

Line 478: It is important to point out that this increase was limited in absolute terms. IFR and IHR at the end of 2022 were still much lower than at the beginning of the pandemic!

REVIEWER COMMENTS

Reviewer #1 (Remarks to the Author):

The manuscript: "The Real Time Infection Hospitalization and Fatality Risk across the COVID-19 Pandemic in the United Kingdom" is well written and provides an overview of the reduction of the Infection and hospitalization fatality rate in the United Kingdom using 2 independent databases. The Method is well described and the conclusions compatible with the results. I suggest some minor corrections:

We thank the reviewer for their time and insightful feedback on the manuscript, which has improved its quality.

1) PCR positivity data show a reduction of positivity time for BA1. The differences in days are small, not being statistically and clinically relevant. This data was not prioritized in the abstract, in the discussion or in the conclusion as the main result, my suggestion is to remove this result from the manuscript and present only the IFR and HFR data. The manuscript is already quite extensive with a long introduction, methodology and results section.

Thank you for this feedback. The results on the length of PCR positivity have been moved to the supplementary section.

2) In the fourth paragraph of introduction: "The 40 earliest estimates found that the two doses of Vaxzevria reduced symptomatic infection by 82.4% 41 (95% CI: 62.7, 91.7) [12] for wild type, dropping to 74.5% (95% CI: 68.4, 79.4) [13] for Alpha, 67.0% (95% CI: 61.3, 71.8) [14] for Delta, and very limited protection was found for Omicron BA.1 [15]. Currently there appears to be no significant difference in the effect of vaccination for Omicron BA.1 and Omicron BA.2 [16] and no current evidence for the effectiveness of two doses of Vaxzevria for Omicron BA.5." Reference 15 only has VE for Delta and not for Omicron. Reference 16 has VE for Omicron BA.1 and not the comparison among BA1 and BA.2. And in the last sentence regarding the BA5 variant, the authors do not quote any reference. It is important to correct the references in the text relating to Vaxzevria.

We thank the reviewer for spotting this error in referencing and this has been corrected.

Reviewer #2 (Remarks to the Author):

I think that this is a very important paper and it has relevance not just for the results but also for methods used to estimate the infection fatality and hospitalisation rates. The manuscript is well written and the presentation of the results is clear. The introduction and discussion are both very clear and I had no major comments on these sections

We thank the reviewer for taking the time to review the manuscript, their feedback, and the comments provided

The key to this method is the prevalence surveys in the UK, from which incidence is estimated, and linking to the hospitalisation and death data through the distributions of the time for infection to event. I am aware of other studies which estimated the IFR through fitting SEIR type models. Are

there other approaches that could be discussed. The discussion of this paper only really discusses the results and the possible reasons for the changes – vaccination, variant, waning protection - and differences among the regions associated with age. I think that there should be a comparison with other estimates of IFR/IHR if there are any such estimates.

We did not find similar methods that were presented in this paper for comparison, which we could cite. However, many past studies used seroprevalence to calculate point estimates (rather than a fortnightly temporal change in the IFR/IHR described here) of the IFR which we have included in the discussion now.

The methods are really very complex and I think that it would be a good idea to try and draw a diagram to outline what you are doing and the stages. You have got daily data on deaths/hospitalisations by date of event and you need to convert this to events among those infected by estimated date of infection. You use the onset to clinical delay distribution to facilitate this back calculation. This gives you the $D(t)$ estimates. You then have the two surveys of prevalence, and you combine them and convert these into incidence estimates. This gives the $I(t)$. You do this explanation in 102-106 but I think a visual display like some flow diagram would help.

We thank the reviewer for this feedback and a visualisation of the methods has now been developed and placed at the front of the methods section.

I was not exactly sure but think that the distinct parts are analysed separately and not all included within a full Bayesian model. I may be wrong here but this is what lines 280-284 suggest to me. I can understand why you do this as the problem is complex but this needs to be clarified.

Due to the distinct environments where certain models are possible to run due to data protection (SRS, UKHSA Halo, Dash, Edge) it was necessary to run certain models separately. The complexity of the problem also makes this more feasible as a stan model that incorporated every component may struggle to complete. The justification for this has now been made clear in the text.

I think that it would be possible to propagate the estimation errors through simulations even if the distinct parts were analysed separately. Also I am not sure if the 5% point of the IFR/IHR simulations based upon the 5% point of the infection estimates is truly the lower limit of a 95% credible interval in a full bayesian analysis.

We thank the reviewer for this feedback and have rerun all the results and changed the approach to uncertainty propagation. Now, we extract the full posterior for key parameters, and provide this posterior as an input to any dependent stan model in the form of a parametric approximation. This allows us to calculate the true credible interval for all parameters of interest and propagates the uncertainty between the different stan models in a parsimonious manner.

In the explanation of the results much is made of the periods when the different variants were circulating suggesting that the changes in the IHR/IFR are associated with variant. While they are there is also going to be an impact of vaccine uptake and waning of protection – particularly for infection, less for serious events – which would have more of an impact on the denominator of the ratios

We agree with the reviewer, but we wanted to contextualise the results with the start of the vaccination campaign and the variants present without making the figures too busy and overcrowded. We have included further text on waning and vaccine uptake to emphasise this point.

Title – this mention the UK but everything in the paper is about England – React was only in England while the ONS survey did have components in Wales, Northern, Ireland and Scotland. I assume you used the English component of the ONS survey. There is no strong reason to suppose that the IHR/IFR would be substantially different in the other nations of the UK but it is better to replace UK in the title with England.

The title has been changed to England.

Minor Points

L91

we only included deaths that had occurred 60 days following a positive RT-PCR test and COVID-19 on the death certificate

What is the time period for this work – Early on the pandemic when testing was limited there were many covid deaths with no PCR test. How have you compensated for this in the analysis?

We have clarified the start dates of the analysis in the paper. The early pandemic is not included within the study period of the analysis. This is because the REACT and ONS studies were too small and the sampling less representative in their initial rounds.

L128

Does the study end on 15/05/22 or are there missing rows.

The analysis included in this study runs until the end of the ONS study (end of March 2023). We have amended this with a further diagram that shows the dates used.

L269-276

It took me a while to work out what you were trying to do here so it would be really helpful to put the full details in the supplementary materials. As I understood you want to combine the two surveys at the same time point. You might consider $(D_O + D_R)/(I_O + I_R)$ as an estimate of theta. The scaling factor is the value you multiply D_R by so that the estimates of I from both studies are the same and so give equal weight. If you didn't do this and I_O and I_R were a bit different then the larger one would dominate and there is no reason to suppose that one survey's estimate of I is better than the others.

We have rewritten this, as it was unclear. The weighting factor we use to weight the two studies is the relative difference in the effective sample size. Therefore, if both studies had the same sample size, we weight the two surveys equally. However, if one study is larger than the other, we assign more weight to that study.

L275

Aren't a and b the shape parameters of the beta distribution. What values did you use for them or did you have hyper priors

We have removed these from the model as they were both set to 1, and were negligible compared to the incidence estimates so had no influence on the model.

L280-284

I understand that you have to take into account the variability in the estimation of I_O and I_R and what you do is reasonable. Is this the 95% credible interval in the paper and graphs. I don't think that this can be a true credible interval – how do you know that the coverage is 95%.

We have now amended the methods in the paper to produce better calibration of the credible intervals. The full posteriors for I_O, I_R are provided to the stan model as inputs, which allows to model to explore the full range of uncertainty in its inputs.

L 292 the CrI limits are the wrong way round – twice

Thank you, this has been corrected.

L288. The credible interval is not a lot wider than you would get with binomial sampling. You don't report the incidence estimates but there are 3500 admissions to hospital at the peak in January 2021 and an IHR of 3.9% suggests 89000 infections with a 95% CI 3.80 to 4.06 which is not much smaller than the reported interval. Given the amount of estimation and calculations I would have expected a wider range. I may be mistaken or the sample sizes just dominate. In the age groups there may be a bigger difference.

Thank you for the comment. Yes, the uncertainty substantially reduces due to the sample size dominating and also greater certainty of the model parameters.

L305-306 For most in the 25-44 age group vaccination did not start until April 2021

Thank you for mentioning this. We plot the lines from when the vaccinations officially became available and were offered for that age group.

L310-319. I thought that this was confusing as you are talking about an 90% increase in IHR from the nadir to now and then you mention a 90% decrease from the peak to now, I guess, as the time is not mentioned. I know that this comes about as the 90% decrease from the peak to nadir is 36 to 1.8 and the 90% increase is from 1.8 to 3.5. I suggest no mentioning the % increases in so much detail and just concentrating on the absolute changes.

Thank you, we agree with the reviewer, and we have removed some of the % changes that might be confusing to a reader.

L328-340

The regional estimates are very interesting as the intervals do not overlap and London seems to have a much lower IHR in Jan 2021. At the time there were a lot of reports about hospitals in London being overwhelmed

The IFR analysis for the regions shows that London has the lowest IFR also – so is this associated with it having a much younger population – Supplementary fig 40 shows 65% in London under 44 and a much smaller % are 75+ - I see you discuss this so nothing extra is needed

Thank you.

L400

I think that the PCR positivity results could also be in the supplementary material. In the order of the analysis you really need this before getting the IHR/IFR so I found it strange to see it at the end of the results.

Thank you for the feedback. This has been moved to the supplementary data section.

L537

I think that the ONS survey does collect personal data and it can be linked to administrative health data within a trusted safe haven.

We agree this would be the ideal approach and the authors did try explicit linkage to clinical data for almost 3 years (the length of ONS CIS) however, this was not possible due to the current wording of the ONS CIS participant agreement. This is something we hope to amend with the ONS participant agreement in the future. However, it is not possible for the data collected hitherto.

Reviewer #3 (Remarks to the Author):

This paper describes the evolution of the risk of hospitalisation and death following SARS-CoV-2 infection during the COVID-19 pandemic in the UK. This important piece of work combines different aggregate data sets using a thorough statistical approach. It allows to describe the evolution of two key indicators during the epidemic. The impact of vaccination and the evolution of the virus are discussed. However, we believe that some significant improvements need to be made before publication. There are four points that we would like to highlight.

We thank the reviewer for their time and insightful feedback.

The methods section should be more explicit about the data used in the model and what they are used for. This should be clear to readers unfamiliar with the organisation of covid-19 surveillance in the UK. It is important to explain why the ONS and REACT data are original (compared to what is available in other countries) and why they made this study possible.

We have provided further context on the studies discussed and the data used in this analysis.

Still in Methods, the reader should be able to better understand the general organisation of the model (what are the different steps and what data are used) without going into the details of the Bayesian models.

Thank you for this feedback. We have provided a flow diagram at the start of the methods section to make the steps easier for the reader to follow.

The methods used are original and advanced. It would be interesting to discuss its advantages and limitations compared to methods based on individual monitoring of infected individuals (e.g. reference [5] or 10.1016/j.eclinm.2022.101455).

Thank you, further discussion has now been included on the advantages and limitations of this approach.

Finally, the English used in the article is quite difficult to read. This applies especially to the title and the discussion. For example, we were puzzled by the following sentences. "Therefore, informed by the time delay distribution analysis, to capture 99% of deaths we included up to 60 days post a positive test or with COVID-19 as the primary cause on the death certificate." " Death certificate confirmed COVID-19 deaths suffer from changes to death reporting practices across the pandemic." Below are some more specific comments.

Thank you for the feedback. We have been through the text and amended any language that could be clearer with specific attention to the points raised below.

The period of time considered in the study is not clearly defined.

This has now been included in the data section of the methods.

It might be interesting to express IR and IFR as cases per 100,000. This would make them easier to read and avoid confusion with the percentage increase or decrease.

We have included the risk of admission or death as an expression of individuals infected, please see the first paragraph of the discussion. The % risk we feel is the most standardised way to present the metric and we have removed relative change where it could be confusing for the reader.

Line 54: When did free testing ceased in the UK?

Free community testing ceased on the 1st April 2022. The ONS study however continued.

Line 84: ONS and REACT 1 should be introduced in more detail here. How are individuals included in these surveys ? Is there a health-seeking bias? Some information is later in the article, such as line 144, but should be grouped here.

We have included more detail on each study and the biases present in the studies

Line 93: NHSE&I is not mentioned as a data source in the summary.

Data sources have been removed from the abstract due to the word limit.

Line 113-115. Did you mean "The first positive date is considered to be the date of the first positive test. The last positive test followed by a negative test is considered as the date of end of infection".

Thank you this has now been amended.

Line 151: The term "round" has not been defined.

Thank you, this has now been amended and defined for each study.

Fig 1: Would it be possible to sort the legend by chronological order of variants?

This has now been corrected in the figures.

Fig 2: Point out that the Y-scales are different for each age group.

This has now been included in the figure descriptions.

Line 396: If I understand correctly, this parameter is used as an input to the model. Therefore, these results should be presented before those of the model or reported in the appendix.

The prevalence studies are presented in the supplementary material.

Line 417: These percentage changes could be presented without decimals. This would make them easier to read.

For consistency of formatting, we have kept decimal places as they otherwise look less standardised in terms of presentation.

Line 418: is "1.2 (1.16, 1.28) in 200" a typo ?

The language has been made clearer in the text.

Line 432: The wording gives the impression that vaccination is one of the causes of the increase!

Thank you, we have amended the language here to clarify the meaning of the sentence.

Line 474: The wording gives the impression that vaccination has not significantly changed the IFR and IHR.

Thank you, we have amended the language here for further clarification.

Line 478: It is important to point out that this increase was limited in absolute terms. IFR and IHR at the end of 2022 were still much lower than at the beginning of the pandemic!

We have changed the language to emphasise this point.

REVIEWERS' COMMENTS

Reviewer #2 (Remarks to the Author):

This version of the paper is much clearer and the authors have addressed all of my queries. I think that the explanation of the methods is much clearer now. It should be possible for a skilled modeller to understand the methods and reproduce this analysis. The methods employed in this paper are complex but have the potential to be very useful in other disease areas.

Reviewer #2 (Remarks on code availability):

There was no code to review. The data are held in safe havens for another researcher would have to apply to run the analyses in these safe havens also. Making the Stan code available for other researchers would probably be a good idea but there are sufficient details in the methods to be able to program up the analysis from the beginning and that might actually be a good test of the method.

Reviewer #3 (Remarks to the Author):

This article has been greatly improved since the first version. In particular, the diagrams now provide a clear overview of the methods and main results.